# HALF-INVERSE GRADIENTS FOR PHYSICAL DEEP LEARNING

**Patrick Schnell, Philipp Holl and Nils Thuerey**
Department of Informatics
Technical University of Munich
Boltzmannstr. 3, 85748 Garching, Germany
{patrick.schnell,philipp.holl,nils.thuerey}@tum.de

## ABSTRACT

Recent works in deep learning have shown that integrating differentiable physics simulators into the training process can greatly improve the quality of results. Although this combination represents a more complex optimization task than supervised neural network training, the same gradient-based optimizers are typically employed to minimize the loss function. However, the integrated physics solvers have a profound effect on the gradient flow as manipulating scales in magnitude and direction is an inherent property of many physical processes. Consequently, the gradient flow is often highly unbalanced and creates an environment in which existing gradient-based optimizers perform poorly. In this work, we analyze the characteristics of both physical and neural network optimizations to derive a new method that does not suffer from this phenomenon. Our method is based on a half-inversion of the Jacobian and combines principles of both classical network and physics optimizers to solve the combined optimization task. Compared to state-of-the-art neural network optimizers, our method converges more quickly and yields better solutions, which we demonstrate on three complex learning problems involving nonlinear oscillators, the Schrödinger equation and the Poisson problem.

## 1 INTRODUCTION

The groundbreaking successes of deep learning (Krizhevsky et al., 2012; Sutskever et al., 2014; Silver et al., 2017) have led to ongoing efforts to study the capabilities of neural networks across all scientific disciplines. In the area of physical simulation, neural networks have been used in various ways, such as creating accurate reduced-order models (Morton et al., 2018), inferring improved discretization stencils (Bar-Sinai et al., 2019), or suppressing numerical errors (Um et al., 2020). The long-term goal of these methods is to exceed classical simulations in terms of accuracy and speed, which has been achieved, e.g., for rigid bodies (de Avila Belbute-Peres et al., 2018), physical inverse problems (Holl et al., 2020), and two-dimensional turbulence (Kochkov et al., 2021).

The successful application of deep learning to physical systems naturally hinges on the training setup. In recent years, the use of physical loss functions has proven beneficial for the training procedure, yielding substantial improvements over purely supervised training approaches (Tompson et al., 2017; Wu & Tegmark, 2019; Greydanus et al., 2019). These improvements were shown to stem from three aspects (Battaglia et al., 2016; Holl et al., 2020): (i) Incorporating prior knowledge from physical principles facilitates the learning process , (ii) the ambiguities of multimodal cases are resolved naturally, and (iii) simulating the physics at training time can provide more realistic data distributions than pre-computed data sets. Approaches for training with physical losses can be divided into two categories. On the one hand, equation-focused approaches that introduce physical residuals (Tompson et al., 2017; Raissi et al., 2019), and on the other hand, solver-focused approaches that additionally integrate well-established numerical procedures into training (Um et al., 2020; Kochkov et al., 2021).

From a mathematical point of view, training a neural network with a physical loss function bears the difficulties of both network training and physics optimization. In order to obtain satisfying

results, it is vital to treat flat regions of the optimization landscapes effectively. In learning, the challenging loss landscapes are addressed using gradient-based optimizers with data-based normalizing schemes, such as Adam (Kingma & Ba, 2015), whereas in physics, the optimizers of choice are higher-order techniques, such as Newton's method (Gill & Murray, 1978), which inherently make use of inversion processes. However, Holl et al. (2021) found that these approaches can not effectively handle the joint optimization of network and physics. Gradient-descent-based optimizers suffer from vanishing or exploding gradients, preventing effective convergence, while higher-order methods do not generally scale to the high-dimensional parameter spaces required by deep learning (Goodfellow et al., 2016).

Inspired by the insight that inversion is crucial for physics problems in learning from Holl et al. (2021), we focus on an inversion-based approach but propose a new method for joint physics and network optimization which we refer to as *half-inverse gradients*. At its core lies a partial matrix inversion, which we derive from the interaction between network and physics both formally and geometrically. An important property of our method is that its runtime scales linearly with the number of network parameters. To demonstrate the wide-ranging and practical applicability of our method, we show that it yields significant improvements in terms of convergence speed and final loss values over existing methods. These improvements are measured both in terms of absolute accuracy as well as wall-clock time. We evaluate a diverse set of physical systems, such as the Schrödinger equation, a nonlinear chain system and the Poisson problem.

## 2 GRADIENTS BASED ON HALF-INVERSE JACOBIANS

Optimization on continuous spaces can be effectively performed with derivative-based methods, the simplest of which is gradient descent. For a target function $L(\boldsymbol{\theta})$ to be minimized of several variables $\boldsymbol{\theta}$, using bold symbols for vector-valued quantities in this section, and learning rate $\eta$, gradient descent proceeds by repeatedly applying updates

$$\Delta\boldsymbol{\theta}_{\mathrm{GD}}(\eta) = -\eta \cdot \left(\frac{\partial L}{\partial \boldsymbol{\theta}}\right)^{\top}. \tag{1}$$

For quadratic objectives, this algorithm convergences linearly with the rate of convergence depending on the condition number $\lambda$ of the Hessian matrix (Lax, 2014). In the ill-conditioned case $\lambda \gg 1$, flat regions in the optimization landscape can significantly slow down the optimization progress. This is a ubiquitous problem in non-convex optimization tasks of the generic form:

$$L(\boldsymbol{\theta}) = \sum_i l\big(\boldsymbol{y}_i(\boldsymbol{\theta}), \hat{\boldsymbol{y}}_i\big) = \sum_i l\big(\boldsymbol{f}(\boldsymbol{x}_i; \boldsymbol{\theta}), \hat{\boldsymbol{y}}_i\big) \tag{2}$$

Here $(\boldsymbol{x}_i, \hat{\boldsymbol{y}}_i)$ denotes the $i$th data points from a chosen set of measurements, $\boldsymbol{f}$ is a function parametrized by $\boldsymbol{\theta}$ to be optimized to model the relationship between the data points $\boldsymbol{y}_i(\boldsymbol{\theta}) = \boldsymbol{f}(\boldsymbol{x}_i; \boldsymbol{\theta})$, and $l$ denotes a loss function measuring the optimization progress. In the following, we assume the most common case of $l(\boldsymbol{y}_i, \hat{\boldsymbol{y}}_i) = \frac{1}{2}||\boldsymbol{y}_i - \hat{\boldsymbol{y}}_i||_2^2$ being the squared L2-loss.

**Physics Optimization.** Simulating a physical system consists of two steps: (i) mathematically modeling the system by a differential equation, and (ii) discretizing its differential operators to obtain a solver for a computer. Optimization tasks occur for instance when manipulating a physical system through an external force to reach a given configuration, for which we have to solve an inverse problem of form 2. In such a control task, the sum reduces to a single data point $(\boldsymbol{x}, \hat{\boldsymbol{y}})$ with $\boldsymbol{x}$ being the initial state, $\hat{\boldsymbol{y}}$ the target state and $\boldsymbol{\theta}$ the external force we want to find. The physical solver corresponds to the function $\boldsymbol{f}$ representing time evolution $\boldsymbol{y}(\boldsymbol{\theta}) = \boldsymbol{f}(\boldsymbol{x}; \boldsymbol{\theta})$. This single data point sum still includes summation over vector components of $\boldsymbol{y} - \hat{\boldsymbol{y}}$ in the L2-loss. Sensitive behavior of the physical system arising from its high-frequency modes is present in the physical solver $\boldsymbol{f}$, and produces small singular values in its Jacobian. This leads to an ill-conditioned Jacobian and flat regions in the optimization landscape when minimizing 2. This is addressed by using methods that incorporate more information than only the gradient. Prominent examples are Newton's method or the Gauss-Newton's algorithm (Gill & Murray, 1978); the latter one is based on the Jacobian of $\boldsymbol{f}$ and the loss gradient:

$$\Delta\boldsymbol{\theta}_{\mathrm{GN}} = -\left(\frac{\partial \boldsymbol{y}}{\partial \boldsymbol{\theta}}\right)^{-1} \cdot \left(\frac{\partial L}{\partial \boldsymbol{y}}\right)^{\top} \tag{3}$$

Here the inversion of the Jacobian is calculated with the pseudoinverse. The Gauss-Newton update maps the steepest descent direction in $\boldsymbol{y}$-space to the parameter space $\boldsymbol{\theta}$. Therefore, to first order, the resulting update approximates gradient descent steps in $\boldsymbol{y}$-space, further details are given in appendix A.2. An advantage of such higher-order methods is that the update steps in $\boldsymbol{y}$-space are invariant under arbitrary rescaling of the parameters $\boldsymbol{\theta}$, which cancels inherent scales in $\boldsymbol{f}$ and ensures quick progress in the optimization landscape.

**Neural Network Training.** For $\boldsymbol{f}$ representing a neural network in equation 2, the optimization matches the typical supervised learning task. In this context, the problem of flat regions in the optimization landscape is also referred to as pathological curvature (Martens, 2010). Solving this problem with higher-order methods is considered to be too expensive given the large number of parameters $\boldsymbol{\theta}$. For learning tasks, popular optimizers, such as Adam, instead use gradient information from earlier update steps, for instance in the form of momentum or adaptive learning rate terms, thereby improving convergence speed at little additional computational cost. Furthermore, the updates are computed on mini-batches instead of the full data set, which saves computational resources and benefits generalization (Goodfellow et al., 2016).

**Neural Network Training with Physics Objectives.** For the remainder of the paper, we consider joint optimization problems, where $\boldsymbol{f}$ denotes a composition of a neural network parameterized by $\boldsymbol{\theta}$ and a physics solver. Using classical network optimizers for minimizing equation 2 is inefficient in this case since data normalization in the network output space is not possible and the classical initialization schemes cannot normalize the effects of the physics solver. As such, they are unsuited to capture the strong coupling between optimization parameters typically encountered in physics applications. While Gauss-Newton seems promising for these cases, the involved Jacobian inversion tends to result in large overshoots in the updates when the involved physics solver is ill-conditioned. As we will demonstrate, this leads to oversaturation of neurons, hampering the learning capability of the neural network.

## 2.1 An Ill-Conditioned Toy Example

To illustrate the argumentation so far, we consider a data set sampled from $\hat{\boldsymbol{y}}(x) = (\sin(6x), \cos(9x))$ for $x \in [-1, 1]$: We train a neural network to describe this data set by using the loss function:

$$l(\boldsymbol{y}, \hat{\boldsymbol{y}}; \gamma) = \frac{1}{2}\left(y^1 - \hat{y}^1\right)^2 + \frac{1}{2}\left(\gamma \cdot y^2 - \hat{y}^2\right)^2 \tag{4}$$

Here, we denote vector components by superscripts. For a scale factor of $\gamma = 1$, we receive the well-conditioned mean squared error loss. However, $l$ becomes increasingly ill-conditioned as $\gamma$ is decreased, imitating the effects of a physics solver. For real-world physics solvers, the situation would be even more complex since these scales usually vary strongly in direction and magnitude across different data points and optimization steps. We use a small neural network with a single hidden layer with 7 neurons and a $\mathrm{tanh}$ activation. We then compare training with the well-conditioned $\gamma = 1$ loss against an ill-conditioned $\gamma = 0.01$ loss. In both cases, we train the network using both Adam and Gauss-Newton as representatives of gradient-based and higher-order optimizers, respectively. The results are shown in figure 1.

In the well-conditioned case, Adam and Gauss-Newton behave similarly, decreasing the loss by about three orders of magnitude. However, in the ill-conditioned case, both optimizers fail to minimize the objective beyond a certain point. To explain this observation, we first illustrate the behavior from the physics viewpoint by considering the trajectory of the network output $\boldsymbol{f}(x)$ for a single value $x$ during training (figure 1, right). For $\gamma = 1$, Adam optimizes the network to accurately predict $\hat{\boldsymbol{y}}(x)$ while for $\gamma = 0.01$, the updates neglect the second component preventing Adam to move efficiently along the small-scale coordinate (blue curve in figure 1b, right). To illustrate the situation from the viewpoint of the network, we consider the variance in the outputs of specific neurons over different $x$ (figure 1, middle). When $\gamma = 1$, all neurons process information by producing different outcomes for different $x$. However, for $\gamma = 0.01$, Gauss-Newton's inversion of the small-scale component $y^2$ results in large updates, leading to an oversaturation of neurons (red curve in figure 1b, middle). These neurons stop processing information, reducing the effective capacity of the network and preventing the network from accurately fitting $\hat{\boldsymbol{y}}$. Facing these problems, a natural questions arises: *Is it possible to construct an algorithm that can successfully process the inherently different scales of a physics solver while training a neural network at the same time?*

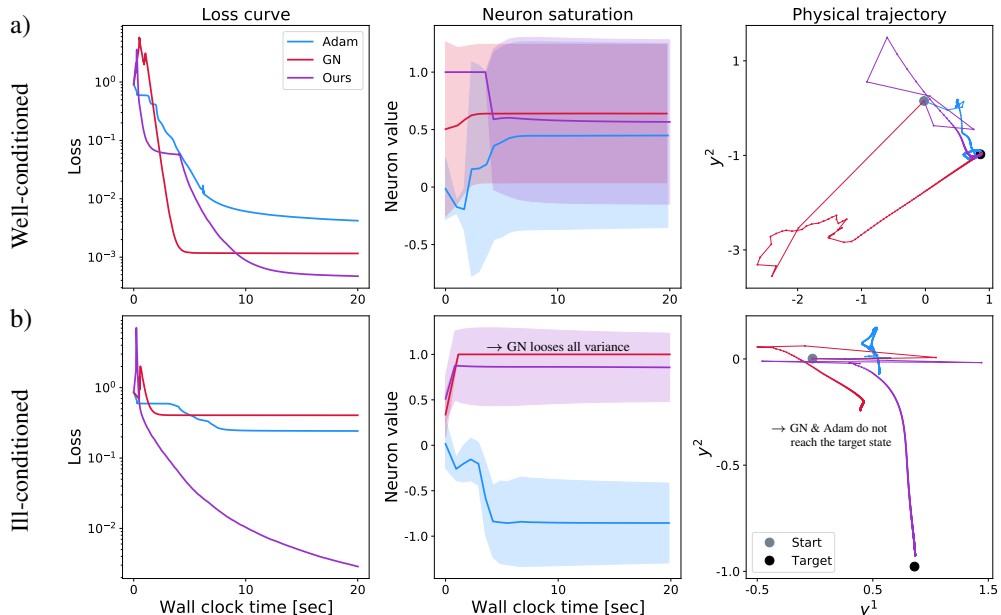

Figure 1: Results of the learning problem of section 2.1. Optimization is performed with a) a well-conditioned loss and b) an ill-conditioned loss. Plots show loss curves over training time (left), data set mean and standard deviation of the output of a neuron output over training time (middle), and the training trajectory of a data point (right).

## 2.2 UPDATES BASED ON HALF-INVERSE JACOBIANS

We propose a novel method for optimizing neural networks with physics objectives. Since pure physics or neural network optimization can be thought of as special cases of the joint optimization, we analogously look for a potential method in the continuum of optimization methods between gradient descent and Gauss-Newton. We consider both of them to be the most elementary algorithms representing network and physics optimizers, respectively. The following equation describes updates that lie between the two.

$$\Delta \boldsymbol{\theta}(\eta, \kappa) = -\eta \cdot \left(\frac{\partial \boldsymbol{y}}{\partial \boldsymbol{\theta}}\right)^{\kappa} \cdot \left(\frac{\partial L}{\partial \boldsymbol{y}}\right)^{\top} \tag{5}$$

Here, the exponent $\kappa$ of the Jacobian denotes the following procedure defined with the aid of the singular value decomposition $J = U \Lambda V^{\top}$:

$$J^{\kappa} := V \Lambda^{\kappa} U^{\top} \tag{6}$$

When $\kappa = 1$, equation 5 reduces to the well-known form of gradient descent. Likewise, the case $\kappa = -1$ yields Gauss-Newton since the result of the Jacobian exponentiation then gives the pseudoinverse of the Jacobian. Unlike other possible interpolations between gradient descent and Gauss-Newton, exponentiation by $\kappa$ as in equation 5 significantly affects the scales inherent in the Jacobian. This is highly important to appropriately influence physics and neural network scales.

To determine $\kappa$, we recall our goal to perform update steps which are optimal in both $\boldsymbol{\theta}$- and $\boldsymbol{y}$-space. However, since any update $\Delta \boldsymbol{\theta}$ and its corresponding effect on the solver output $\Delta \boldsymbol{y}$ are connected by the inherent scales encoded in the Jacobian, no single $\kappa$ exists that normalizes both at the same time. Instead, we distribute the burden equally between network and physics by choosing $\kappa = -1/2$. From a geometric viewpoint, the resulting update can be regarded as a steepest descent step when the norm to measure distance is chosen accordingly. This alternative way to approach our method is explained in the appendix (A.2) and summarized in table 1.

For batch size $b$ and learning rate $\eta$, we define the following update step for our method by stacking network-solver Jacobians $\frac{\partial \boldsymbol{y}_i}{\partial \boldsymbol{\theta}}\big|_{\boldsymbol{x}_i}$ and loss gradients $\frac{\partial L}{\partial \boldsymbol{y}_i}\big|_{\boldsymbol{x}_i, \hat{\boldsymbol{y}}_i}$ of different data points $(\boldsymbol{x}_i, \hat{\boldsymbol{y}}_i)$:

Table 1: Optimization algorithms viewed as steepest descent algorithm w.r.t. the given L2-norms.

| Optimization Method | performs Steepest Descent: | Norm ($\boldsymbol{\theta}$-space) | | Norm ($\boldsymbol{y}$-space) |
|---|---|---|---|---|
| **Gradient Descent** | in Parameter Space | $\|\cdot\|_{\boldsymbol{\theta}}$ | $=$ | $\|\cdot\|_{J^{-1}\boldsymbol{y}}$ |
| **Gauss-Newton** | in Physics Space | $\|\cdot\|_{J\boldsymbol{\theta}}$ | $=$ | $\|\cdot\|_{\boldsymbol{y}}$ |
| **Ours** | in Intermediate Space | $\|\cdot\|_{J^{3/4}\boldsymbol{\theta}}$ | $=$ | $\|\cdot\|_{J^{-1/4}\boldsymbol{y}}$ |

$$
\Delta\boldsymbol{\theta}_{\text{HIG}} = -\eta \cdot \begin{pmatrix} \frac{\partial \boldsymbol{y}_1}{\partial \boldsymbol{\theta}}\big|_{\boldsymbol{x}_1} \\ \frac{\partial \boldsymbol{y}_2}{\partial \boldsymbol{\theta}}\big|_{\boldsymbol{x}_2} \\ \vdots \\ \frac{\partial \boldsymbol{y}_b}{\partial \boldsymbol{\theta}}\big|_{\boldsymbol{x}_b} \end{pmatrix}^{-1/2} \cdot \begin{pmatrix} \frac{\partial L}{\partial \boldsymbol{y}_1}\big|_{\boldsymbol{x}_1,\hat{\boldsymbol{y}}_1}^{\top} \\ \frac{\partial L}{\partial \boldsymbol{y}_2}\big|_{\boldsymbol{x}_2,\hat{\boldsymbol{y}}_2}^{\top} \\ \vdots \\ \frac{\partial L}{\partial \boldsymbol{y}_b}\big|_{\boldsymbol{x}_b,\hat{\boldsymbol{y}}_b}^{\top} \end{pmatrix} \tag{7}
$$

Besides batch size $b$ and learning rate $\eta$, we specify a truncation parameter $\tau$ as an additional hyperparameter enabling us to suppress numerical noise during the half-inversion process in equation 6. As with the computation of the pseudoinverse via SVD, we set the result of the $-\frac{1}{2}$-exponentiation of every singular value smaller than $\tau$ to 0.

The use of a *half-inversion* – instead of a full inversion – helps to prevent exploding updates of network parameters while still guaranteeing substantial progress in directions of low curvature. With the procedure outlined above, we arrived at a balanced method that combines the advantages of optimization methods from deep learning and physics. As our method uses half-inverse Jacobians multiplied with gradients we refer to them in short as *half-inverse gradients* (HIGs).

**Half-inverse Gradients in the Toy Example.** With the definition of HIGs, we optimize the toy example introduced in section 2.1. The results in figure 1 show that for $\gamma = 1$, HIGs minimize the objective as well as Adam and Gauss-Newton's method. More interestingly, HIGs achieve a better result than the other two methods for $\gamma = 0.01$. On the one hand, the physics trajectory (figure 1b, right) highlights that HIGs can process information along the small-scale component $y^2$ well and successfully progress along this direction. On the other hand, by checking neuron saturation (figure 1b, middle), we see that HIGs – in contrast to Gauss Newton – avoid oversaturating neurons.

## 2.3 PRACTICAL CONSIDERATIONS

**Computational Cost.** A HIG update step consists of constructing the stacked Jacobian and computing the half-inversion. The first step can be efficiently parallelized on modern GPUs, and therefore induces a runtime cost comparable to regular backpropagation at the expense of higher memory requirements. In situations where the computational cost of the HIG step is dominated by the half-inversion, memory requirements can be further reduced by parallelizing the Jacobian computation only partially. At the heart of the half-inversion lies a divide and conquer algorithm for the singular value decomposition (Trefethen & Bau, 1997). Hence, the cost of a HIG step scales as $\mathcal{O}(|\boldsymbol{\theta}| \cdot b^2 \cdot |\boldsymbol{y}|^2)$, i.e. is linear in the number of network parameters $|\boldsymbol{\theta}|$, and quadratic in the batch size $b$ and the dimension of the physical state $|\boldsymbol{y}|$. Concrete numbers for memory requirements and duration of a HIG step are listed in the appendix.

**Hyperparameters.** Our method depends on several hyperparameters. First, we need a suitable choice of the learning rate. The normalizing effects of HIGs allow for larger learning rates than commonly used gradient descent variants. We are able to use $\eta = 1$ for many of our experiments. Second, the batch size $b$ affects the number of data points included in the half-inversion process. It should be noted that the way the feedback of individual data points is processed is fundamentally different from the standard gradient optimizers: Instead of the averaging procedure of individual gradients of a mini batch, our approach constructs an update that is optimal for the complete batch. Consequently, the quality of updates increases with higher batch size. However, overly large batch sizes can cause the Jacobian to become increasingly ill-conditioned and destabilize the learning progress. In appendix C, we discuss the remaining parameters $\tau$ and $\kappa$ with several ablation experiments to illustrate their effects in detail.

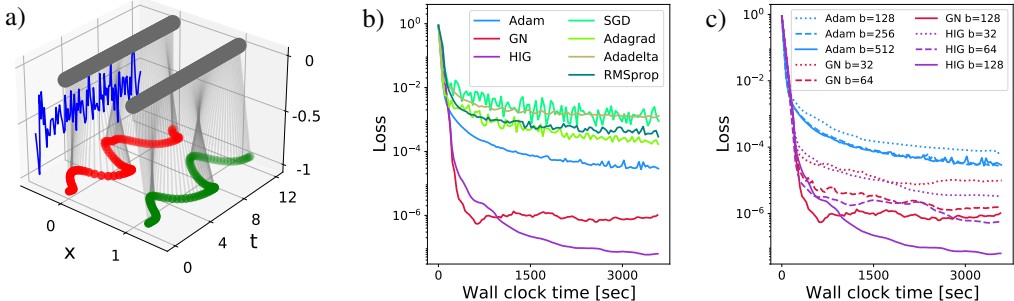

Figure 2: Nonlinear oscillator system: a) Time evolution controlled by a HIG-trained neural network. Its inferred output is shown in blue in the background. b) Loss curves for different optimization methods. c) Loss curves for Adam, GN, and HIG with different batch sizes $b$.

## 3 EXPERIMENTS

We evaluate our method on three physical systems: controlling nonlinear oscillators, the Poisson problem, and the quantum dipole problem. Details of the numerical setups are given in the appendix along with results for a broad range of hyperparameters. For a fair comparison, we show results with the best set of hyperparameters for each of the methods below and plot the loss against wall clock time measured in seconds. All learning curves are recorded on a previously unseen data set.

### 3.1 CONTROL OF NONLINEAR OSCILLATORS

First, we consider a control task for a system of coupled oscillators with a nonlinear interaction term. This system is of practical importance in many areas of physics, such as solid state physics (Ibach & Lüth, 2003). Its equations of motions are governed by the Hamiltonian

$$\mathcal{H}(x_i, p_i, t) = \sum_i \left( \frac{x_i^2}{2} + \frac{p_i^2}{2} + \alpha \cdot (x_i - x_{i+1})^4 + u(t) \cdot x_i \cdot c_i \right), \tag{8}$$

where $x_i$ and $p_i$ denote the Hamiltonian conjugate variables of oscillator $i$, $\alpha$ the interaction strength, and the vector $c$ specifies how to scalar-valued control function $u(t)$ is applied. In our setup, we train a neural network to learn the control signal $u(t)$ that transforms a given initial state into a given target state with 96 time steps integrated by a 4th order Runge-Kutta scheme. We use a dense neural network with three hidden layers totalling 2956 trainable parameters and ReLU activations. The Mean-Squared-Error loss is used to quantify differences between predicted and target state. A visualization of this control task is shown in figure 2a.

**Optimizer comparison.** The goal of our first experiments is to give a broad comparison of the proposed HIGs with commonly used optimizers. This includes stochastic gradient descent (SGD), Adagrad (Duchi et al., 2011), Adadelta (Zeiler, 2012), RMSprop (Hinton et al., 2012), Adam (Kingma & Ba, 2015), and Gauss-Newton (GN) applied to mini batches. The results are shown in figure 2b where all curves show the best runs for each optimizer with suitable hyperparameters independently selected, as explained in the appendix. We find that the state-of-the-art optimizers stagnate early, with Adam achieving the best result with a final loss value of $10^{-4}$. In comparison, our method and GN converge faster, exceeding Adam's accuracy after about three minutes. While GN exhibits stability problems, the best stable run from our hyperparameter search reaches a loss value of $10^{-6}$. HIGs, on the other hand, yield the best result with a loss value of $10^{-7}$. These results clearly show the potential of our method to process different scales of the physics solver more accurately and robustly. They also make clear that the poor result of the widely-used network optimizers cannot be attributed to simple numerical issues as HIG converges to better levels of accuracy with an otherwise identical setup.

**Role of the batch size.** We conduct multiple experiments using different values for the batch size $b$ as a central parameter of our method. The results are shown in figure 2c. We observe that for

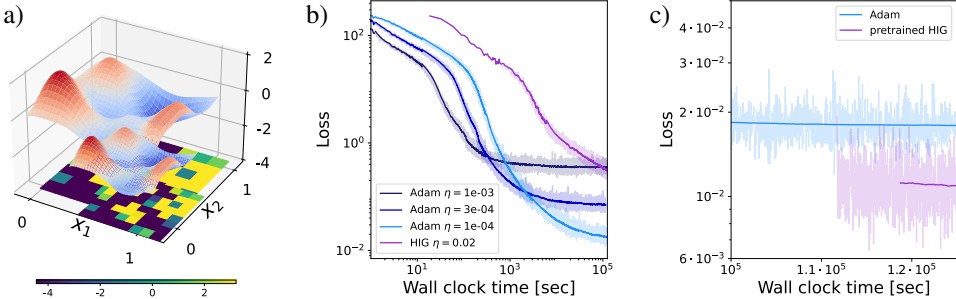

Figure 3: Poisson problem: a) Example of a source distribution $\rho$ (bottom) and inferred potential field (top). b) Loss curves of Adam and HIG training for different learning rates $\eta$. c) Loss curves of Adam ($\eta = 0.0001$), and HIG ($\eta = 0.02$) pretrained with Adam.

Adam, all runs converge about equally quickly while HIGs and GN show improvements from larger batch sizes. This illustrates an important difference between Adam and HIG: Adam uses an average of gradients of data points in the mini batch, which approaches its expectation for large $b$. Further increasing the batch size has little influence on the updates. In contrast, our method includes the individual data point gradients without averaging. As shown in equation 7, we construct updates that are optimized for the whole batch by solving a linear system. This gives our method the ability to hit target states very accurately with increasing batch size. To provide further insights into the workings of HIGs, we focus on detailed comparisons with Adam as the most popular gradient descent variant.

## 3.2 POISSON PROBLEM

Next we consider Poisson's equation to illustrate advantages and current limitations of HIGs. Poisson problems play an important role in electrostatics, Newtonian gravity, and fluid dynamics (Ames, 2014). For a source distribution $\rho(x)$, the goal is to find the corresponding potential field $\phi(x)$ fulfilling the following differential equation:

$$\Delta\phi = \rho \tag{9}$$

Classically, Poisson problems are solved by solving the corresponding system of linear equations on the chosen grid resolution. Instead, we train a dense neural network with three hidden layers and 41408 trainable parameters to solve the Poisson problem for a given right hand side $\rho$. We consider a two-dimensional system with a spatial discretization of $8 \times 8$ degrees of freedom. An example distribution and solution for the potential field are shown in figure 3a.

**Convergence and Runtime.** Figure 3b shows learning curves for different learning rates when training the network with Adam and HIGs. As we consider a two-dimensional system, this optimization task is challenging for both methods and requires longer training runs. We find that both Adam and HIGs are able to minimize the loss by up to three orders of magnitude. The performance of Adam varies, and its two runs with larger $\eta$ quickly slow down. In terms of absolute convergence per time, the Adam curve with the smallest $\eta$ shows advantages in this scenario. However, choosing a log-scale for the time axis reveals that both methods have not fully converged. In particular, while the Adam curve begins to flatten at the end, the slope of the HIG curve remains constant and decreases with a steeper slope than Adam. The performance of Adam can be explained by two reasons. First, the time to compute a single Adam update is much smaller than for HIGs, which requires the SVD solve from equation 6. While these could potentially be sped up with appropriate methods (Foster et al., 2011; Allen-Zhu & Li, 2016), the absolute convergence per iteration, shown in the appendix in figure 7, shows how much each HIG update improves over Adam. Second, compared to the other examples, the Poisson problem is relatively simple, requiring only a single matrix inversion. This represents a level of difficulty which Adam is still able to handle relatively well.

**HIGs with Adam Pretraining.** To further investigate the potential of HIGs, we repeat the training, this time using the best Adam model from figure 3b for network initialization. While Adam progresses slowly, HIGs are able to quickly improve the state of the neural network, resulting in

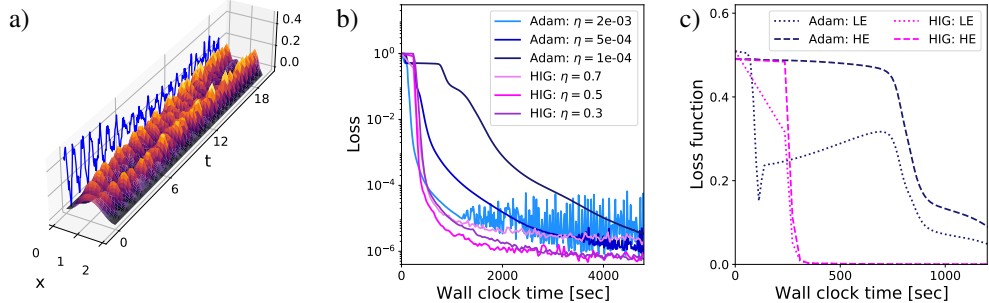

Figure 4: Quantum dipole: a) A transition of two quantum states in terms of probability amplitude $|\Psi(t, x)|^2$, controlled by a HIG-trained neural network. Its inferred output is shown in blue in the background. b) Loss curves with Adam and HIGs for different $\eta$. c) Low-energy (LE) and high-energy (HE) loss with Adam ($\eta = 0.0001$) and HIG ($\eta = 0.5$).

a significant drop of the loss values, followed by a faster descent than Adam. Interestingly, this experiment indicates that the HIG updates are able to improve aspects of the solution which Adam is agnostic to. Despite outlining the potential gains from faster SVD calculations, this example also highlights the quality of the HIG updates for simpler PDEs.

### 3.3 QUANTUM DIPOLE

As a final example, we target the quantum dipole problem, a standard control task formulated on the Schrödinger equation and highly relevant in quantum physics (Von Neumann, 2018). Given an initial and a target state, we train a neural network to compute the temporal transition function $u(t)$ in an infinite-well potential $V$ according the evolution equation of the physical state $\Psi$:

$$i\partial_t \Psi = \big( -\Delta + V + u(t) \cdot \hat{x} \big)\Psi \qquad (10)$$

We employ a modified Crank-Nicolson scheme (Winckel et al., 2009) for the discretization of spatial and temporal derivatives. Thus, each training iteration consists of multiple implicit time integration steps – 384 in our setup – for the forward as well as the backward pass of each mini-batch. The control task consists of inferring a signal that converts the ground state to a given randomized linear combination of the first and the second excited state. We use a dense neural network with three hidden layers, $9484$ trainable parameters and $\tanh$ activations. Similarity in quantum theories is quantified with inner products; therefore, our loss function is given by $L(\Psi_a, \Psi_b) = 1 - |\langle \Psi_a, \Psi_b \rangle|^2$. A visualization of this control task is shown in figure 4a.

**Speed and Accuracy.** We observe that HIGs minimize the loss faster and reach a better final level of accuracy than Adam (figure 4b). While the Adam run with the largest learning rate drops faster initially, its final performance is worse than all other runs. In this example, the difference between the final loss values is not as large as for the previous experiments. This is due to the numerical accuracy achievable by a pure physics optimization, which for our choice of parameters is around $10^{-6}$. Hence, we can not expect to improve beyond this lower bound for derived learning problems. Our results indicate that the partial inversion of the Jacobian successfully leads to the observed improvements in convergence speed and accuracy.

**Low and High Energy Components.** The quantum control problem also serves to highlight the weakness of gradient-based optimizers in appropriately processing different scales of the solutions. In the initial training stage, the Adam curves stagnate at a loss value of $0.5$. This is most pronounced for $\eta = 10^{-4}$ in dark blue. To explain this effect, we recall that our learning objective targets transitions to combinations of the 1st and 2nd excited quantum states, and both states appear on average with equal weight in the training data. Transitions to the energetically higher states are more difficult and connected to smaller scales in the physics solver, causing Adam to fit the lower-energetic component first. In contrast, our method is constructed to process small scales in the Jacobian via the half-inversion more efficiently. As a consequence, the loss curves decrease faster below $0.5$. We support this explanation by explicitly plotting separate loss curves in figure 4c

quantifying how well the low and high energy component of the target state was learned. Not only does Adam prefer to minimize the low-energy loss, it also increases the same loss again before it is able to minimize the high-energy loss. In contrast, we observe that HIGs minimize both losses uniformly. This is another indication for the correctness of the theory outlined above of an more even processing of different scales in joint physics and neural network objectives through our method.

## 4  RELATED WORK

**Optimization algorithms.**  Optimization on continuous spaces is a huge field that offers a vast range of techniques (Ye et al., 2019). Famous examples are gradient descent (Curry, 1944), Gauss-Newton's method (Gill & Murray, 1978), Conjugate Gradient (Hestenes et al., 1952), or the limited-memory BFGS algorithm (Liu & Nocedal, 1989). In deep learning, the preferred methods instead rely on first order information in the form of the gradient, such as SGD (Bottou, 2010) and RMSProp (Hinton et al., 2012). Several methods approximate the diagonal of the Hessian to improve scaling behavior, such as Adagrad (Duchi et al., 2011), Adadelta (Zeiler, 2012), and most prominently, Adam (Kingma & Ba, 2015). However, due to neglecting interdependencies of parameters, these methods are limited in their capabilities to handle physical learning objectives. Despite the computational cost, higher-order methods have also been studied in deep learning (Pascanu & Bengio, 2013) . Practical methods have been suggested by using a Kroenecker-factorization of the Fisher matrix (Martens & Grosse, 2015), iterative linear solvers (Martens, 2010), or by recursive approximations of the Hessian (Botev et al., 2017). To the best of our knowledge, the only other technique specifically targeting optimization of neural networks with physics objectives is the inversion approach from Holl et al. (2021). However, their updates are based on inverse physics solvers, while we address the problem by treating network and solver as an entity and half-inverting its Jacobian. Thus, we work on the level of linear approximations while updates based on physics inversion are able to harness higher-order information provided that an higher-order inverse solver exists. Additionally, they compute their update by averaging gradients over different data points, in line with typical gradient-based neural network optimizers. HIGs instead process the feedback of different data points via collective inversion.

**Incorporating physics.**  Many works involve differentiable formulations of physical models, e.g., for robotics (Toussaint et al., 2018), to enable deep architectures (Chen et al., 2018), as a means for scene understanding (Battaglia et al., 2013; Santoro et al., 2017), or the control of rigid body environments de Avila Belbute-Peres et al. (2018). Additional works have shown the advantages of physical loss formulations (Greydanus et al., 2019; Cranmer et al., 2020). Differentiable simulation methods were proposed for a variety of phenomena, e.g. for fluids (Schenck & Fox, 2018), PDE discretizations (Bar-Sinai et al., 2019), molecular dynamics (Wang et al., 2020), reducing numerical errors (Um et al., 2020), and cloth (Liang et al., 2019; Rasheed et al., 2020). It is worth noting that none of these works question the use of standard deep learning optimizers, such as Adam. In addition, by now a variety of specialized software frameworks are available to realize efficient implementations (Hu et al., 2020; Schoenholz & Cubuk, 2019; Holl et al., 2020).

## 5  DISCUSSION AND OUTLOOK

We have considered optimization problems of neural networks in combination with physical solvers and questioned the current practice of using the standard gradient-based network optimizers for training. Derived from an analysis of smooth transitions between gradient descent and Gauss-Newton's method, our novel method learns physics modes more efficiently without overly straining the network through large weight updates, leading to a faster and more accurate minimization of the learning objective. This was demonstrated with a range of experiments.

We believe that our work provides a starting point for further research into improved learning methods for physical problems. Highly interesting avenues for future work are efficient methods for the half-inversion of the Jacobian matrix, or applying HIGs to physical systems exhibiting chaotic behavior or to more sophisticated training setups (Battaglia et al., 2013; Ummenhofer et al., 2020; Pfaff et al., 2020).

ACKNOWLEDGEMENTS

This work was supported by the ERC Consolidator Grant CoG-2019-863850 *SpaTe*, and by the DFG SFB-Transregio 109 *DGD*. We would also like to express our gratitude to the reviewers and the area chair for their helpful feedback.

REPRODUCIBILITY STATEMENT

Our code for the experiments presented in this paper is publicly available at `https://github.com/tum-pbs/half-inverse-gradients`. Additionally, the chosen hyperparameters are listed in the appendix along with the hardware used to run our simulations.

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

# APPENDIX

## A    FURTHER DETAILS ON OPTIMIZATION ALGORITHMS

Our work considers optimization algorithms for functions of the form $\boldsymbol{f}(\boldsymbol{x};\boldsymbol{\theta}) = \boldsymbol{y}$ with $\boldsymbol{\theta}, \Delta\boldsymbol{\theta} \in \mathbb{R}^t$, denoting weight vector and weight update vector, respectively, while $\boldsymbol{x} \in \mathbb{R}^n$ and $\boldsymbol{y} \in \mathbb{R}^m$ denote input and output. The learning process solves the minimization problem $\mathrm{argmin}_{\boldsymbol{\theta}} L(\boldsymbol{f}(\boldsymbol{x};\boldsymbol{\theta}), \hat{\boldsymbol{y}})$ via a sequence $\boldsymbol{\theta}^{k+1} = \boldsymbol{\theta}^k + \eta\Delta\boldsymbol{\theta}$. Here, $\hat{\boldsymbol{y}}$ are the reference solutions, and we target losses of the form $L(\boldsymbol{x}, \hat{\boldsymbol{y}}; \boldsymbol{\theta}) = \sum_i l(\boldsymbol{f}(\boldsymbol{x}_i; \boldsymbol{\theta}), \hat{\boldsymbol{y}}_i)$ with $i$ being an index for multiple data points (i.e., observations). $l$ denotes the L2-loss $\sum_j ||\boldsymbol{x}_j - \hat{\boldsymbol{y}}_j||^2$ with $j$ referencing the entries of a mini batch of size $b$.

### A.1    UPDATE STEP OF THE GAUSS-NEWTON ALGORITHM

Using this notation, the update step of the Gauss-Newton algorithm (Adby, 2013) for $\eta = 1$ is given by:

$$\Delta\boldsymbol{\theta}_{\mathrm{GN}} = -\left(\left(\frac{\partial\boldsymbol{y}}{\partial\boldsymbol{\theta}}\right)^T \cdot \left(\frac{\partial\boldsymbol{y}}{\partial\boldsymbol{\theta}}\right)\right)^{-1} \cdot \left(\frac{\partial\boldsymbol{y}}{\partial\boldsymbol{\theta}}\right)^T \cdot \left(\frac{\partial L}{\partial\boldsymbol{y}}\right)^{\top} \tag{11}$$

The size of the Jacobian matrix is given by the dimensions of $\boldsymbol{y}$- and $\boldsymbol{\theta}$-space. For a full-rank Jacobian corresponding to non-constrained optimization, the Gauss-Newton update is equivalent to:

$$\Delta\boldsymbol{\theta}_{\mathrm{GN}} = -\left(\frac{\partial\boldsymbol{y}}{\partial\boldsymbol{\theta}}\right)^{-1} \cdot \left(\frac{\partial L}{\partial\boldsymbol{y}}\right)^{\top} \tag{12}$$

Even in a constrained setting, we can reparametrize the coordinates to obtain an unconstrained optimization problem on the accessible manifold and rewrite $\Delta\boldsymbol{\theta}_{\mathrm{GN}}$ similarly. This shortened form of the update step is given in equation 3, and is the basis for our discussion in the main text.

### A.2    GEOMETRIC INTERPRETATION AS STEEPEST DESCENT ALGORITHMS

It is well-known that the negative gradient of a function $L(\boldsymbol{\theta})$ points in the direction of steepest descent leading to the interpretation of gradient descent as a steepest descent algorithm. However, the notion of steepest descent requires defining a measure of distance, which is in this case the usual L2-norm in $\boldsymbol{\theta}$. By using different metrics, we can regard Gauss-Newton and HIG steps as steepest descent algorithms as well.

**Gauss-Newton updates.**    The updates $\Delta\boldsymbol{\theta}_{\mathrm{GN}}$ can be regarded as gradient descent in $\boldsymbol{y}$ up to first order in the update step. This can be seen with a simple equation by considering how these updates change $\boldsymbol{y}$.

$$\Delta\boldsymbol{y} = \left(\frac{\partial\boldsymbol{y}}{\partial\boldsymbol{\theta}}\right) \cdot \Delta\boldsymbol{\theta}_{\mathrm{GN}} + o(\Delta\boldsymbol{\theta}_{\mathrm{GN}}) = -\left(\frac{\partial L}{\partial\boldsymbol{y}}\right)^{\top} + o(\Delta\boldsymbol{\theta}_{\mathrm{GN}}) \tag{13}$$

In figure 1 of the main paper, this property is visible in the physics trajectories for the well-conditioned case, where $L(\boldsymbol{y})$ is a uniform L2-loss and hence, gradient descent in $\boldsymbol{y}$ produces a straight line to the target point. The Gauss-Newton curve first shows several steps in varying directions as the higher-order terms from the neural network cannot be neglected yet. However, after this initial phase the curve exhibits the expected linear motion.

The behavior of GN to perform steepest descent on the $\boldsymbol{y}$-manifold stands in contrast to gradient descent methods, which instead perform steepest descent on the $\boldsymbol{\theta}$-manifold. This geometric view is the basis for an alternative way to derive our method that is presented below.

**HIG updates.** HIG updates can be regarded as a steepest descent algorithm, again up to first order in the update step, when measuring distances of $\boldsymbol{\theta}$-vectors with the following semi-norm:

$$||\boldsymbol{\theta}||_{HIG} := ||J^{3/4}\boldsymbol{\theta}|| \tag{14}$$

Here $|| \cdot ||$ denotes the usual L2-norm and $J = \frac{\partial \boldsymbol{y}}{\partial \boldsymbol{\theta}}$ the Jacobian of network and solver. The exponentiation is performed as explained in the main text, with $J = U\Lambda V^\top$ being the SVD, and $J^{3/4}$ given by $V\Lambda^{3/4}U^\top$. Additionally, we will use the natural map between dual vector and vector $\langle \cdot, \cdot \rangle$ and the loss gradient $g = \frac{\partial L}{\partial \boldsymbol{y}}$.

To prove the claim above, we expand the loss around an arbitrary starting point $\boldsymbol{\theta}_0$:

$$L(y(\boldsymbol{\theta}_0 + \Delta\boldsymbol{\theta})) = L(y(\boldsymbol{\theta}_0)) + \langle g \cdot J, \Delta\boldsymbol{\theta} \rangle + o(\Delta\boldsymbol{\theta}) \tag{15}$$

The first term on the right-hand side is constant and the third term is neglected according to the assumptions of the claim. Hence, we investigate for which fixed-length $\Delta\boldsymbol{\theta}$ the second term decreases the most:

$$
\begin{aligned}
\underset{||\Delta\boldsymbol{\theta}||_{HIG}=const.}{\arg\min} \left( \langle g \cdot J, \Delta\boldsymbol{\theta} \rangle \right) &= \underset{||\boldsymbol{\theta}||_{HIG}=const.}{\arg\min} \left( \langle g \cdot J^{1/4}, J^{3/4}\Delta\boldsymbol{\theta} \rangle \right) \\
&= \underset{\gamma}{\arg\min} \left( \cos\gamma \cdot \underbrace{||g \cdot J^{1/4}||}_{const.} \cdot \underbrace{||J^{3/4}\Delta\boldsymbol{\theta}||}_{=const.} \right) \\
&= \underset{\gamma}{\arg\min} \left( \cos\gamma \right)
\end{aligned}
\tag{16}
$$

In the first step above, we split the Jacobian $J^\top = V\Lambda U^\top = (V\Lambda^{1/4}V^\top)(V\Lambda^{3/4}U^\top) = J^{1/4}J^{3/4}$. $\gamma$ denotes the angle between $J^{1/4}g^\top$ and $J^{3/4}\Delta\boldsymbol{\theta}$. This expression is minimized for $\gamma = -\pi$, meaning the two vectors have to be antiparallel:

$$J^{3/4}\Delta\boldsymbol{\theta} = -J^{1/4}g^\top \tag{17}$$

This requirement is fulfilled by the HIG update $\Delta\boldsymbol{\theta}_{HIG} = -J^{1/2}g^\top$, and is therefore a steepest descent method, which concludes our proof.

This presents another approach to view HIGs as an interpolation between gradient descent and Gauss-Newton's method. More precisely, gradient descent performs steepest descent in the usual L2-norm in $\boldsymbol{\theta}$-space ($||\boldsymbol{\theta}||$). Considering only terms up to linear order, Gauss-Newton performs steepest descent in the L2-norm in $\boldsymbol{y}$-space ($||J\boldsymbol{\theta}||$). The HIG update ($||J^{3/4}\boldsymbol{\theta}||$) lies between these two methods. The quarter factors in the exponents result from the additional factor of 2 that has to be compensated for when considering L2-norms.

## A.3 STABILITY OF INVERSIONS IN THE CONTEXT OF PHYSICAL DEEP LEARNING.

In the following, we illustrate how the full inversion of GN can lead to instabilities at training time. Interestingly, physical solvers are not the only cause of small singular values in the Jacobian. They can also occur when applying equation 12 to a mini batch to train a neural network and are not caused by numerical issues. Consider the simple case of two data points $(\boldsymbol{x}_1, \hat{\boldsymbol{y}}_1)$ and $(\boldsymbol{x}_2, \hat{\boldsymbol{y}}_2)$ and a one-dimensional output. Let $f$ be the neural network and $J$ the Jacobian, which is in this case the gradient of the network output. Then equation 12 yields:

$$
\begin{pmatrix} J_{\boldsymbol{f}}(\boldsymbol{x}_1) \\ J_{\boldsymbol{f}}(\boldsymbol{x}_2) \end{pmatrix} \cdot \Delta\boldsymbol{\theta}_{\text{GN}} = \begin{pmatrix} \boldsymbol{f}(\boldsymbol{x}_1) - \hat{\boldsymbol{y}}_1 \\ \boldsymbol{f}(\boldsymbol{x}_2) - \hat{\boldsymbol{y}}_2 \end{pmatrix}
\tag{18}
$$

Next, we linearly approximate the second row by using the Hessian $H$ by assuming the function to be learned is $\hat{\boldsymbol{f}}$, i.e. $\hat{\boldsymbol{f}}(\boldsymbol{x}_1) = \boldsymbol{y}_1$ and $\hat{\boldsymbol{f}}(\boldsymbol{x}_2) = \boldsymbol{y}_2$. Neglecting terms beyond the linear approximation, we receive:

$$\begin{pmatrix} J_{\boldsymbol{f}}(\boldsymbol{x}_1) \\ J_{\boldsymbol{f}}(\boldsymbol{x}_1) + H_{\boldsymbol{f}}(\boldsymbol{x}_1) \cdot (\boldsymbol{x}_2 - \boldsymbol{x}_1) \end{pmatrix} \cdot \Delta \boldsymbol{\theta}_{\mathrm{GN}} = \begin{pmatrix} \boldsymbol{f}(\boldsymbol{x}_1) - \boldsymbol{y}_1 \\ \boldsymbol{f}(\boldsymbol{x}_1) - \boldsymbol{y}_1 + (J_{\boldsymbol{f}}(\boldsymbol{x}_1) - J_{\hat{\boldsymbol{f}}}(\boldsymbol{x}_1)) \cdot (\boldsymbol{x}_2 - \boldsymbol{x}_1) \end{pmatrix}$$
(19)

Considering the case of two nearby data points, i.e. $\boldsymbol{x}_2 - \boldsymbol{x}_1$ being small, the two row vectors in the stacked Jacobian on the left-hand side are similar, i.e. the angle between them is small. This leads to a small singular value of the stacked Jacobian. In the limit of $\boldsymbol{x}_2 = \boldsymbol{x}_1$ both row vectors are linearly dependant and hence, one singular value becomes zero.

Moreover, even if $\boldsymbol{x}_2$ is not close to $\boldsymbol{x}_1$, small singular values can occur if the batch size increases: for a growing number of row vectors it becomes more and more likely that the Jacobian contains similar or linearly dependent vectors.

After inversion, a small singular value becomes large. This leads to a large update $\Delta \boldsymbol{\theta}_{\mathrm{GN}}$ when the right-hand side of equation 19 overlaps with the corresponding singular vector.

This can easily happen if the linear approximation of the right-hand side is poor, for instance when $\hat{\boldsymbol{f}}$ is a solution to an inverse physics problem. Then $\hat{\boldsymbol{f}}$ can have multiple modes and can, even within a mode, exhibit highly sensitive or even singular behavior.

In turn, applying large updates to the network weights naturally can lead to the oversaturation of neurons, as illustrated above, and diverging training runs in general.

As illustrated in the main paper, these inherent problems of GN are alleviated by the partial inversion of the HIG. It yields a fundamentally different order of scaling via its square-root inversion, which likewise does not guarantee that small singular values lead to overshoots (hence the truncation), but in general strongly stabilizes the training process.

## B  EXPERIMENTAL DETAILS

In the following, we provide details of the physical simulations used for our experiments in section 3 of the main paper. For the different methods, we use the following abbreviations: half-inverse gradients (HIG), Gauss-Newton's method (GN), and stochastic gradient descent (GD). Learning rates are denoted by $\eta$, batch sizes by $b$, and truncation parameters for HIG and GN by $\tau$. All loss results are given for the average loss over a test set with samples distinct from the training data set.

For each method, we run a hyperparameter search for every experiment, varying the learning rate by several orders of magnitude, and the batch size in factors of two. Unless noted otherwise, the best runs in terms of final test loss were selected and shown in the main text. The following sections contain several examples from the hyperparameter search to illustrate how the different methods react to the changed settings.

**Runtime Measurements**   Runtimes for the non-linear chain and quantum dipole were measured on a machine with Intel Xeon 6240 CPUs and NVIDIA GeForce RTX 2080 Ti GPUs. The Poisson experiments used an Intel Xeon W-2235 CPU with NVIDIA Quadro RTX 8000 GPU. We experimentally verified that these platforms yield an on-par performance for our implementation. As deep learning API we used TensorFlow version 2.5. If not stated otherwise, each experiment retained the default settings.

All runtime graphs in the main paper and appendix contain wall-clock measurements that include all steps of a learning run, such as initialization, in addition to the evaluation time of each epoch. However, the evaluations of the test sets to determine the performance in terms of loss are not included. As optimizers such as Adam typically performs a larger number of update steps including these evaluations would have put these optimizers at an unnecessary disadvantage.

### B.1  TOY EXAMPLE (SECTION 2.1)

For the toy example, the target function is given by $\hat{f}(x) = (\sin(6x), \cos(9x))$. We used a dense neural network consisting of one hidden layer with 7 neurons and $\tanh$ activation, and an output layer with 2 neurons and linear activation. For training, we use 1024 data points uniformly sampled

Table 2: Hyperparameters for different optimization algorithms in figure 2b

| Method | Adadelta | Adagrad | Adam | GN | HIG | RMSprop | SGD |
|--------|----------|---------|------|-----|-----|---------|-----|
| $b$ | 512 | 512 | 512 | 128 | 128 | 512 | 512 |
| $\eta$ | 0.1 | 0.1 | $3 \cdot 10^{-4}$ | - | 1 | $10^{-4}$ | 0.1 |
| $\tau$ | - | - | - | $10^{-6}$ | $10^{-6}$ | - | - |

Table 3: Nonlinear oscillators: memory requirements, update duration and duration of the Jacobian computation for Adam and HIG

| Optimizer | Adam | Adam | Adam | HIG | HIG | HIG |
|-----------|------|------|------|-----|-----|-----|
| Batch size | 256 | 512 | 1024 | 32 | 64 | 128 |
| Memory (MB) | 11.1 | 22.2 | 44.5 | 169 | 676 | 2640 |
| Update duration (sec) | 0.081 | 0.081 | 0.081 | 0.087 | 0.097 | 0.146 |
| Jacobian duration (sec) | 0.070 | 0.070 | 0.070 | 0.070 | 0.070 | 0.070 |

from the $[-1, 1]$ interval, and a batch size of 256. For the optimizers, the following hyperparameters were used for both the well-conditioned loss and the ill-conditioned loss: Adam $\eta = 0.3$; GN has no learning rate (equivalent to $\eta = 1$), $\tau = 10^{-4}$; HIG $\eta = 1.0$, $\tau = 10^{-6}$.

## B.2 CONTROL OF NONLINEAR OSCILLATORS (SECTION 3.1)

The Hamiltonian function given in equation 8 leads to the following equations of motions:

$$\ddot{x}_i = -x_i + 4\alpha(x_i - x_{i-1})^3 - 4\alpha(x_i - x_{i+1})^3 - u(t) \cdot c_i \tag{20}$$

The simulations of the nonlinear oscillators were performed for two mass points and a time interval of 12 units with a time step $\Delta t = 0.125$. This results in 96 time steps via 4th order Runge-Kutta per learning iteration. We generated 4096 data points for a control vector $c = (0.0, 3.0)$, and an interaction strength $\alpha = 1.0$ with randomized conjugate variables $x$ and $p$. The test set consists of 4096 new data points. For the neural network, we set up a fully-connected network with ReLU activations passing inputs through three hidden layers with 20 neurons in each layer before being mapped to a 96 output layer with linear activation.

For the comparison with other optimizers (figure 2b) we performed a broad hyperparameter search for each method, as outlined above, to determine suitable settings. The parameters for Adagrad (Duchi et al., 2011), Adadelta (Zeiler, 2012), Adam (Kingma & Ba, 2015), RMSprop (Hinton et al., 2012), Gauss-Newton (Gill & Murray, 1978), HIGs, and stochastic gradient descent (Curry, 1944) are summarized in table 2. For figure 2c the following hyperparameters were used: $\eta = 3 \cdot 10^{-4}$ for Adam, and $\eta = 1.0$, $\tau = 10^{-6}$ for HIG.

**Further Experiments.** Figure 5 and figure 6 contain additional runs with different hyperparameters for the method comparison of figure 2b in the main paper. The graphs illustrate that all five method do not change their behavior significantly for the different batch sizes in each plot, but become noticeably unstable for larger learning rates $\eta$ (plots on the right sides of each section).

Details on the memory footprint and update durations can be found in table 3. Since our simulations were not limited by memory, we used an implementation for the Jacobian computation of HIGs, which scales quadratically in the batch size. Should this become a bottleneck, this scaling could potentially be made linear by exploiting that the Jacobian of the physical solver for multiple data points is blockdiagonal.

## B.3 POISSON PROBLEM (SECTION 3.2)

We discretize Poisson's equation on a regular grid for a two-dimensional domain $\Omega = [0, 8] \times [0, 8]$ with a grid spacing of $\Delta x = 1$. Dirichlet boundary conditions of $\phi = 0$ are imposed on all four sides of $\Omega$. The Laplace operator is discretized with a finite difference stencil (Ames, 2014).

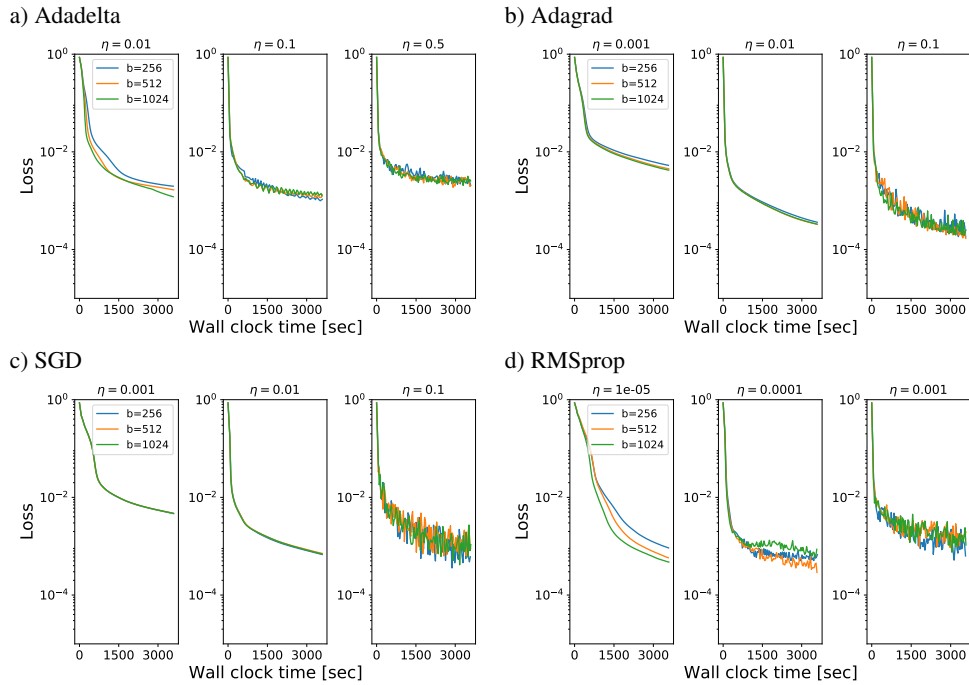

Figure 5: Control of nonlinear oscillators: Additional experiments with (a) Adadelta, (b) Adagrad, (c) stochastic gradient descent , and (d) RMSprop. Each showing different learning rates $\eta$ and batch sizes $b$.

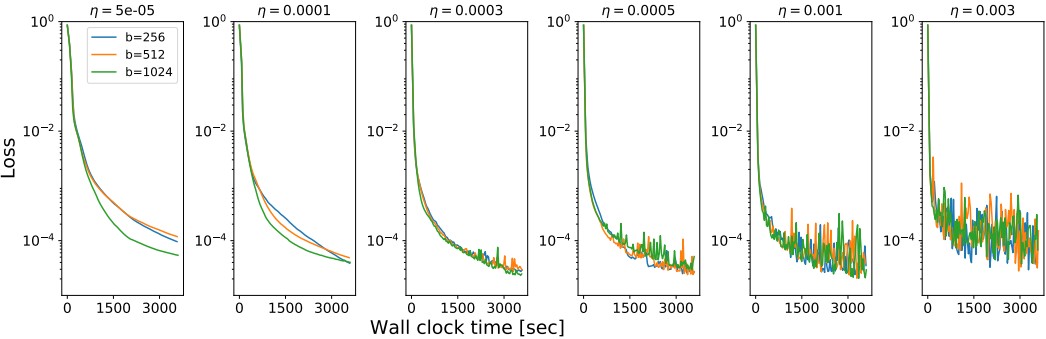

Figure 6: Control of nonlinear oscillators: Additional experiments with Adam for different learning rates $\eta$ and batch sizes $b$.

For the neural network, we set up a fully-connected network with $\tanh$ activation functions. The 8x8 inputs pass through three hidden layers with 64, 256 and 64 neurons, respectively, before being mapped to 8x8 in the output layer. For training, source distributions $\rho$ are sampled from random frequencies in Fourier space, and transformed to real space via the inverse Fourier transform. The mean value is normalized to zero. We sample data on-the-fly, resulting in an effectively infinite data set. This makes a separate test set redundant as all training data is previously unseen.

**Further Experiments.** Figure 7a shows Adam and HIG runs from figure 3b over epochs. The HIG runs converge faster per iteration, which indicates that HIGs perform qualitatively better updates.

Additionally, we use the pretrained HIG run from figure 3c as a starting point for further Adam training. The results are shown in 7b. We observe that the network quickly looses the progress the HIGs have made, and continues with a loss value similar to the orginal Adam run. This again

Table 4: Poisson problem: memory requirements, update duration and duration of the Jacobian computation for Adam and HIG

| Optimizer | Adam | HIG |
|---|---|---|
| Batch size | 64 | 64 |
| Memory (MB) | 1.3 | 3560 |
| Update duration (sec) | 0.011 | 13.8 |
| Jacobian duration (sec) | 0.010 | 0.0035 |

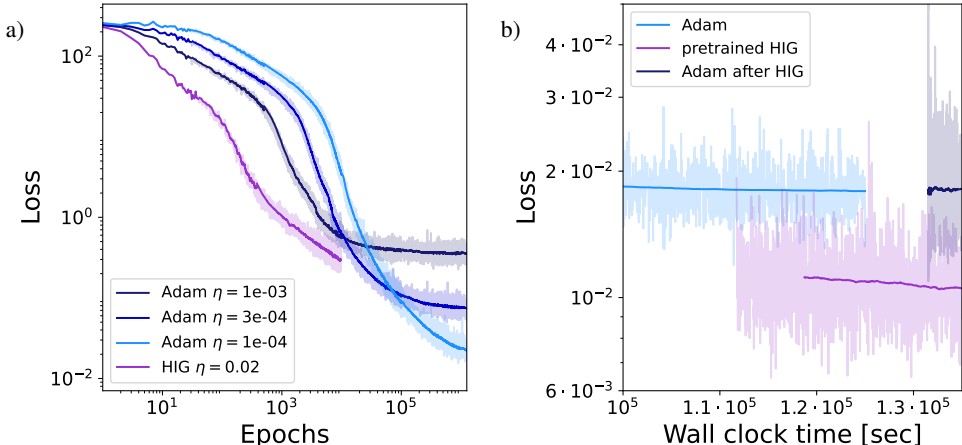

Figure 7: Poisson problem: a) Loss curves for Adam and HIG per epoch for different learning rates, b) Loss curves of Adam ($\eta$ =1e-04), of HIG ($\eta = 0.02$) pretrained with Adam, and of Adam ($\eta$ =1e-04) pretrained with the HIGs.

supports our intuition that Adam, in contrast to HIGs, cannot harness the full potential of the physics solver.

Details on the memory footprint and update durations can be found in table 4

### B.4 QUANTUM DIPOLE (SECTION 3.3)

For the quantum dipole problem, we discretize the Schrödinger equation on a spatial domain $\Omega = [0, 2]$ with a spacing of $\Delta x = 0.133$ resulting in 16 discretization points. We simulate up to a time of 19.2 with a time step of $\Delta t = 0.05$, which yields 384 time steps. Spatial and temporal discretization use a modified Crank-Nicolson scheme (Winckel et al., 2009) which is tailored to quantum simulations. The training data set consists of 1024 randomized superpositions of the first and second excited state, while the test set contains a new set of 1024 randomized superpositions. For the neural network, we set up a fully-connected network with $\tanh$ activations passing the inputs through three hidden layers with 20 neurons in each layer before being mapped to a 384 neuron output layer with linear activation. Overall, the network contains 9484 trainable parameters.

**Experimental details.** For the training runs in figure 4b, Adam used $b = 256$, while for HIG $b = 16$, and $\tau = 10^{-5}$ were used. For the training runs in figure 4c, Adam used $b = 256$, $\eta = 0.0001$, while HIGs used $b = 16$, $\tau = 10^{-5}$, and $\eta = 0.5$. Details on the memory footprint and update durations can be found in table 5

Figure 8 and figure 9 show the performance of both methods for a broader range of $\tau$ settings for HIGs, and $\eta$ for Adam. For Adam, a trade-off between slow convergence and oscillating updates exists. The HIGs yield high accuracy in training across a wide range of values for $\tau$, ranging from $10^{-5}$ to $10^{-3}$. This supports the argumentation in the main text that the truncation is not

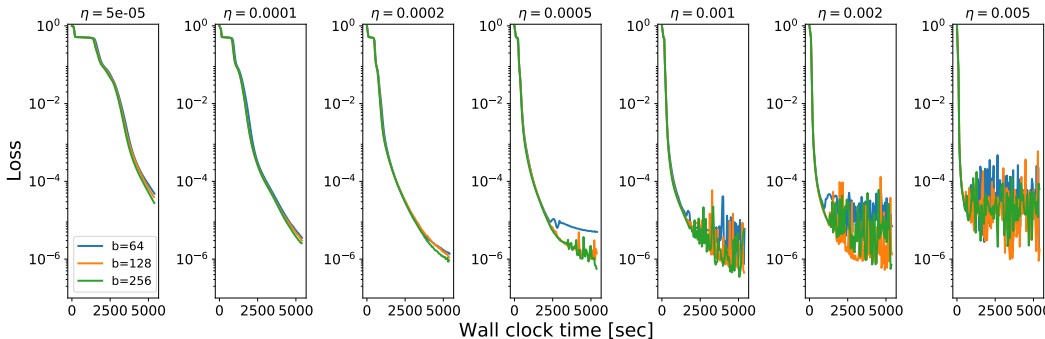

Figure 8: Quantum dipole: Additional experiments with Adam for different learning rates $\eta$ and batch sizes $b$.

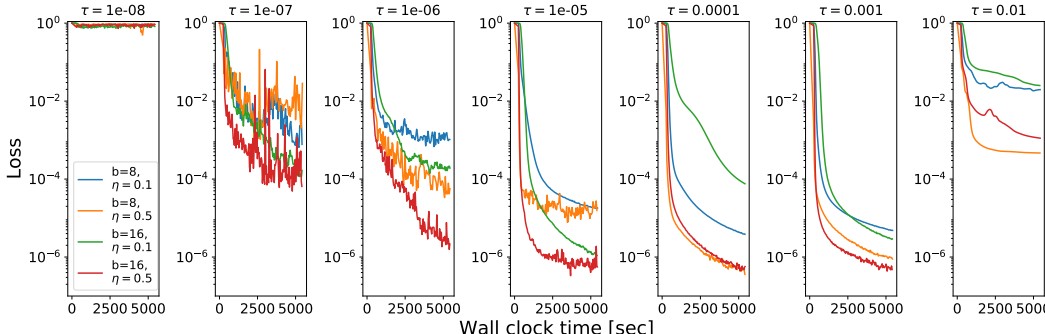

Figure 9: Quantum dipole: Additional experiments with HIGs for different learning rates $\eta$, batch sizes $b$, and truncation parameters $\tau$

overly critical for HIGs. As long as numerical noise is suppressed with $\tau > 10^{-6}$, and the actual information about scaling of network parameters and physical variables is not cut off. The latter case is visible for an overly large $\tau = 0.01$ in the last graph on the right.

Note that many graphs in figure 9 contain a small plateau at the start of each training run. These regions with relatively small progress per wall clock time are caused by the initialization overhead of the underlying deep learning framework (TensorFlow in our case). As all graphs measure wall clock time, we include the initialization overhead of TensorFlow, which causes a noticeable slow down of the first iteration. Hence, the relatively slow convergence of the very first steps in figure 9 are not caused by conceptual issues with the HIGs themselves. Rather, they are a result of the software frameworks and could, e.g., be alleviated with a pre-compilation of the training graphs. In contrast, the initial convergence plateaus of Adam with smaller $\eta$ in Figure 8 are of a fundamentally different nature: they are caused by an inherent problem of non-inverting optimizers: their inability to appropriately handle the combination of large and small scale components in the physics of the quantum dipole setup (as outlined in section 3.3).

Table 5: Quantum dipole: memory requirements, update duration and duration of the Jacobian computation for Adam and HIG

| Optimizer | Adam | Adam | Adam | HIG | HIG |
|---|---|---|---|---|---|
| Batch size | 256 | 512 | 1024 | 8 | 16 |
| Memory (MB) | 460 | 947 | 2007 | 1064 | 5039 |
| Update duration (sec) | 0.40 | 0.50 | 1.33 | 0.42 | 0.60 |
| Jacobian duration (sec) | 0.39 | 0.49 | 1.32 | 0.40 | 0.53 |

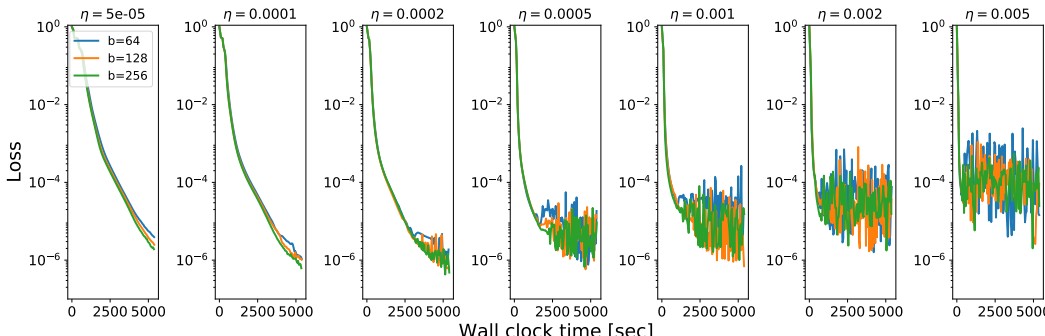

Figure 10: Quantum dipole with Convolutional Neural Network: Experiments with Adam for different learning rates $\eta$ and batch sizes $b$.

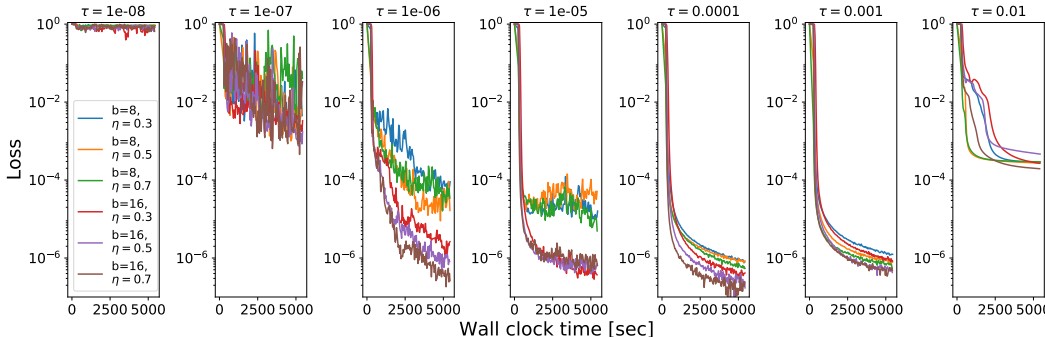

Figure 11: Quantum dipole with Convolutional Neural Network: Experiments with HIGs for different learning rates $\eta$, batch sizes $b$, and truncation parameters $\tau$

**Loss Functions.** While training is evaluated in terms of the regular inner product as loss function: $L(\Psi_a, \Psi_b) = 1 - |\langle \Psi_a, \Psi_b \rangle|^2$, we use the following modified losses to evaluate low- and high-energy states for figure 4c. Let $\Psi_1$ be the first excited state, then we define the low-energy loss as:

$$L(\Psi_a, \Psi_b) = (|\langle \Psi_a, \Psi_1 \rangle| - |\langle \Psi_1, \Psi_b \rangle|)^2$$

Correspondingly, we define the high-energy loss with the second excited state $\Psi_2$:

$$L(\Psi_a, \Psi_b) = (|\langle \Psi_a, \Psi_2 \rangle| - |\langle \Psi_2, \Psi_b \rangle|)^2$$

**Additional Experiments with a Convolutional Neural Network.** Our method is agnostic to specific network architectures. To illustrate this, we conduct additional experiments with a convolutional neural network. The setup is the same as before, only the fully-connected neural network is replaced by a network with 6 hidden convolutional layers each with kernel size 3, 20 features and `tanh` activation, followed by an 384 neuron dense output layer with linear activation giving the network a total of 21984 trainable parameters.

The results of these experiments are plotted in figure 10 and 11. We find that HIGs behave in line with the fully-connected network case (figure 9). There exists a range $\tau$-values from around $10^{-5}$ to $10^{-3}$ for which stable training is possible. Regarding optimization with Adam, we likewise observe a faster and more accurate minimization of the loss function for the best HIG run ($\eta = 0.7$, $b = 16$, $\tau = 10^{-4}$) compared to the best Adam run ($\eta = 0.0002$, $b = 256$).

# C ABLATION STUDY

In this last section, we investigate how the HIG-hyperparameters affect the outcome. This includes ablation experiments with respect to $\kappa$ and $\tau$ defined in section 2.2. We use the nonlinear oscillator example as the basis for these comparisons and consider the following HIG update step:

$$\Delta\boldsymbol{\theta}(\eta,\beta,\kappa) = -\eta \cdot \left(\frac{\partial \boldsymbol{y}}{\partial \boldsymbol{\theta}}\right)^{<\beta,\kappa>} \cdot \left(\frac{\partial L}{\partial \boldsymbol{y}}\right)^{\top} \tag{21}$$

Here, the exponent $< \beta, \kappa >$ of the Jacobian denotes the following procedure defined with the aid of the singular value decomposition $J = U\Lambda V^{\top}$ as:

$$J^{<\beta,\kappa>} := \max\{\mathrm{diag}(\Lambda)\}^{\beta} \cdot V\Lambda^{\kappa}U^{\top}, \tag{22}$$

Compared to the HIG update 5 in the main text, update 21 has an additional scalar prefactor with an parameter $\beta$ resulting from earlier experiments with our method. Setting $\beta = -1 - \kappa$ yields algorithms that rescale the largest singular value to 1, which ensures that the resulting updates cannot produce arbitrarily large updates in $y$-space. This can be thought of as a weaker form of scale invariance. Just as 5, equation 21 defines an interpolation between gradient descent ($\beta = 0$, $\kappa = 1$) and the Gauss-Newton method ($\beta = 0$, $\kappa = -1$) as well.

**Scalar prefactor term $\beta$:** We test $\beta$-values between 0, no scale correction, and $-0.5$, which fully normalizes the effect of the largest singular value for $\kappa = -0.5$. The results are shown in figure 12a. Compared to the other hyperparameters, we observe that $\beta$ has only little influence on the outcome, which is why we decided to present the method without this parameter in the main text.

**Exponent of the diagonal singular value matrix $\kappa$:** We test $\kappa$ for various values between 1.0, stochastic gradient descent, and $-1$, Gauss-Newton. The results are shown in figure 12b. For positive values, curves stagnate early, while for negative $\kappa$, the final loss values are several orders of magnitude better. The HIG curve corresponding to $\beta = -0.5$ achieves the best result. This supports our argumentation that a strong dependence on this parameter exists, and that a choice of $\kappa = -0.5$ is indeed a good compromise for scale-correcting updates of reasonable size. The strong improvement as soon as $\kappa$ becomes negative indicates that the collective inversion of the feedback of different data points of the mini-batch is an important ingredient in our method.

**Truncation parameter $\tau$:** To understand the effect of this parameter, we consider the singular value decomposition (SVD) of the network-solver Jacobian, which is determined by the SVDs of the network Jacobian and the solver Jacobian. The singular values of a matrix product AB depend non-trivially on the singular values of the matrices A and B. In the simplest case, the singular values of the matrix product are received by multiplication of the individual singular values of both matrix factors. In the general case, this depends on how the singular vectors of A and B overlap with each other. However, it is likely that singular vectors with a small singular value of A or B overlap significantly with singular vectors with a small singular value of AB. For this reason, it is important not to truncate too much as this might remove the small-scale physics modes that we are ultimately trying to preserve in order to achieve accurate results. On the other hand, less truncation leads to large updates of network weights on a scale beyond the validation of the linear approximation by first-order derivatives. These uncontrolled network modifications can lead to over-saturated neurons and prevent further training progress.

From a practical point of view, we choose $\tau$ according to the accuracy of the pure physics optimization problem without a neural network. For the quantum dipole training, this value was set to $10^{-5}$. Trying to solve the pure physics optimization with far smaller values leads to a worse result or no convergence at all. The network training behaves in line with this: Figure 9 shows that the network does not learn to control the quantum system with $\tau$-values far smaller than $10^{-5}$. For the nonlinear oscillator system, the pure physics optimization is stable over a large range of $\tau$-values with similarly good results. For the network training, we chose $\tau$ to be $10^{-6}$. We conducted further experiments for the network training with different $\tau$ from $10^{-5}$ to $10^{-10}$ presented in figure 13,

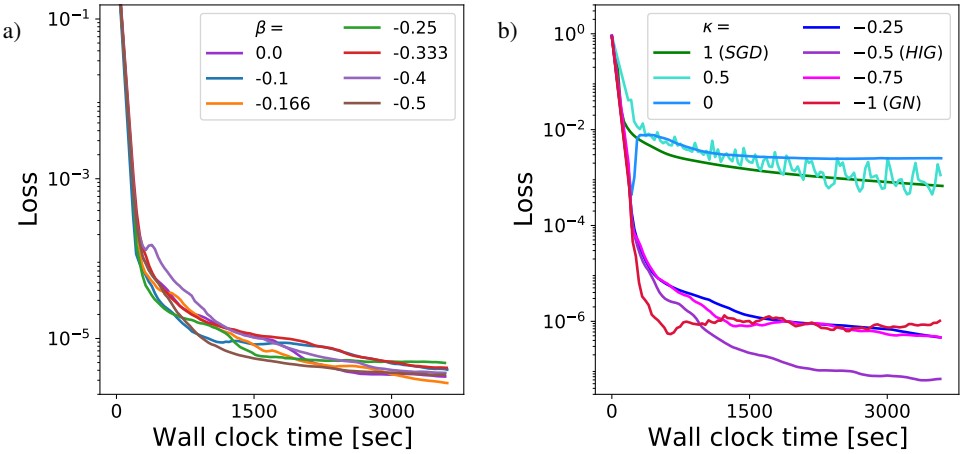

Figure 12: a) Ablation experiments with the $\beta$-hyperparameter, and b) with the $\kappa$-hyperparameter.

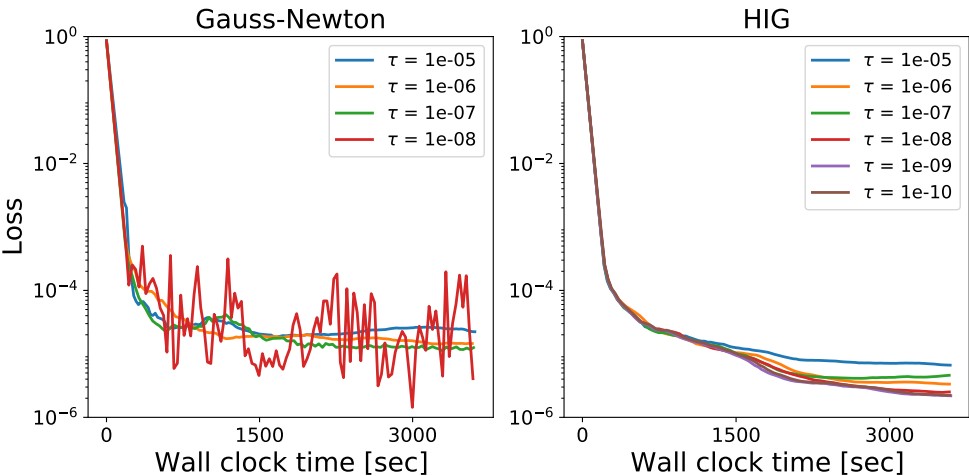

Figure 13: Ablation experiments with the $\tau$-hyperparameter.

which show that HIGs have a similar tolerance in $\tau$. For a comparison, we also plotted Gauss-Newton curves for different $\tau$. We observe that GN curves become more unstable for smaller truncation values $\tau$ and diverge in the case $10^{-9}$ and $10^{-10}$ while HIG curves achieve overall better loss values and start to converge in this parameter.

