# OpenReview forum: "Half-Inverse Gradients for Physical Deep Learning"
_ICLR.cc/2022/Conference — ICLR 2022 Spotlight_

### Official Review · Reviewer_itKt · 2021-11-01

**Correctness:** 3
**Technical Novelty And Significance:** 3
**Empirical Novelty And Significance:** 3
**Recommendation:** 6
**Confidence:** 3

**Details Of Ethics Concerns:**

The paper does not have the ethics concerns.

**Main Review:**

Strengths:
The paper presents an efficient method for training neural networks for physical simulations. The idea is simple and is easy to understand. The written and organization of paper is clear and easy to follow.

Weaknesses:
I have some concerns on the experiments.
1. It is better to compare the batch size of HIGs with the Gauss-Newton method.
2. The truncation threshold \tau is important for both accuracy and efficiency. It is better to discuss its effect in more details. Also, how to set \tau for a problem?
3. What does \gamma refer to throughout our the paper?
4. Just Below Eq.(2), "f" should be "$f$".

**Summary Of The Paper:**

The paper considers the training of neural networks for physical simulations. By distributing the burden equally between network and physics, the paper presents the half-inverse gradients (HIGs) method. Experiments show its advantage for achieving a faster and more accurate minimization.

**Summary Of The Review:**

The paper is well-written and organized. The idea is simple and effective. It is better to discuss more details on the settings of the hyper-parameters in the proposed method.

---

> ### Author Response · Authors · 2021-11-17
> **Response to Reviewer itKt**
>
> Dear reviewer,
>
> Thank you for your review. We are happy to address your concerns and illuminate some of the more practical aspects of our method.
>
> * **Comparing batch size for half-inverse gradients (HIGs) and Gauss-Newton (GN)**
> We agree that it is a good idea to compare batch sizes between HIGs and GN, since they are more directly related than those between HIGs and Adam. The derivatives computed for individual data points are processed in different ways namely averaging for the well-established gradient methods, and collective inversion for HIGs (and GN). In our original submission, we focused on HIGs and Adam to illustrate this important difference between both methods.
> Nevertheless, we are happy to include GN experiments in the batch size plot (figure 2c). Comparing GN vs HIGs for fixed batch sizes, the behavior is similar to the batch size 128 case shown in the paper (figure 2b). The Gauss-Newton curve for a batch size of 32 shows larger loss values and is less smooth than the corresponding HIG curve. The same holds for a batch size of 64.
> Comparing batch sizes for GN, the 32 curve shows larger loss values than the 64 curve, and the 64 curve shows larger loss values than the 128 curve. Final loss values for the GN/HIG curves, ordered from worst to best, are summarized below:
> | Method        | Batch size           |   Final loss value  |
> | ------------- |:-------------:|:-----:|
> | GN      | 32 | $9.9\cdot 10^{-6}$ |
> | HIG      | 32     |   $3.3\cdot 10^{-6}$ |
> | GN | 64      |    $1.6\cdot 10^{-6}$ |
> | GN      | 128 | $1.0 \cdot 10^{-6}$ |
> | HIG      | 64     |  $5.7\cdot 10^{-7}$ |
> | HIG | 128      |    $6.3 \cdot 10^{-8}$ |
>
> * **Truncation parameter $\tau$**
> The singular value decomposition (SVD) of the network-solver Jacobian is determined by the SVDs of the network Jacobian and the solver Jacobian. The singular values of a matrix product AB depend non-trivially on the singular values of the matrices A and B. In the simplest case, the singular values of the matrix product are received by multiplication of the individual singular values of both matrix factors. In the general case, this depends on how the singular vectors of A and B overlap with each other. However, it is likely that singular vectors with a small singular value of A or B overlap significantly with singular vectors with a small singular value of AB. For this reason, it is important not to truncate too much as this might remove the small-scale physics modes that we are ultimately trying to preserve in order to achieve accurate results.
> On the other hand, less truncation leads to large updates of network weights on a scale beyond the validation of the linear approximation by first-order derivatives. These uncontrolled network modifications can lead to oversaturated neurons and prevent further training progress.
> From a practical point of view, we chose $\tau$ according to the accuracy of the pure physics optimization problem without a neural network. For the quantum dipole training, this value was set to $10^{-5}$. Trying to solve the pure physics optimization with far smaller values leads to a worse result or no convergence at all. The network training behaves in line with this: Figure 10 in the appendix shows that the network does not learn to control the quantum system with $\tau$-values far smaller than $10^{-5}$. For the nonlinear oscillator, the pure physics optimization is stable over a large range of $\tau$-values with similarly good results. For the network training, we chose $\tau$ to be $10^{-6}$. We conducted further experiments for the network training with different taus from $10^{-6}$ to $10^{-10}$. They show that HIGs have a similar tolerance in $\tau$. The HIG curves start to converge when lowering the truncation, with final loss values as follows:
> | Method        | Truncation parameter $\tau$          |   Final loss value  |
> | ------------- |:-------------:|:-----:|
> | HIG     | $10^{-6}$ | $3.3\cdot 10^{-6}$ |
> | HIG      | $10^{-8}$     |   $2.5\cdot 10^{-6}$ |
> | HIG | $10^{-10}$      |     $2.2\cdot 10^{-6}$|
>
>   In comparison, the final loss values of GN curves are above $10^{-5}$, with highly oscillating behavior over two orders of magnitude for $\tau=10^{-8}$. For $\tau=10^{-10}$, the loss even explodes. We will include this discussion and results in the paper to illustrate the behavior of the $\tau$-hyperparameter.
>
> * **Meaning of $\gamma$**
> In the loss of the toy example, $\gamma$ is the scale parameter, which we unfortunately denoted as $\lambda$ in the loss definition. We apologize for the confusion. Lastly, we will also update our manuscript with respect to the suggestions of the other reviewers about notation, formulations and argumentation. In case there are more statements that are confusing or need further clarification, we are happy to make additional changes.

---

### Official Review · Reviewer_Khvk · 2021-11-02

**Correctness:** 4
**Technical Novelty And Significance:** 4
**Empirical Novelty And Significance:** 3
**Recommendation:** 8
**Confidence:** 3

**Main Review:**

I personally feel excited about this direction. Data-driven physical learning has been a hot area recently where progress is being made in multiple parallel paths including problem formulation, network architectures, loss specifications, etc. The optimization paradigm has been an unavoidable component in deep learning, and this work fits into the place as the analogy of its counterparts in vision/NLP problems. This manuscript has shown convincing theoretical intuition and validated on valuable physical problems, nevertheless, it arouses my curiosity in several directions:

1. I found this work strongly parallel with a recent pre-print [1], which I would like to refer to as a concurrent work to this manuscript. I am well aware that a comparison is not required but would like to hear more qualitative discussion.
2. It seems to me that the loss function in the toy example should be $l(y,\hat{y},\lambda)=\frac{1}{2}(y^1-\hat{y}^1)^2+\frac{1}{2}\lambda(y^2-\hat{y}^2)^2$, or did I miss something? Also the notation (superscripts and subscripts) are confusing.
3. The computational cost seems high. The Jacobian inference, in spite of being vectorized for a subset of modules, can be a memory monster. I expect some results and discussions on the memory footprint. Also, I wonder if the method can benefit from the Krylov subspace method where an iterative method is used for speedup.
4. My experience with Adam tells me learning rate scheduler is a good friend of first-order methods. I wonder if one is used for Adam baselines in the experiments. If not, how does it perform?
5. A very common trick in training neural networks is adding normalization layers (in recurrent cases, layer normalization is more popular). How does this method deal with normalization layers? Does it require it? If it does, how to update it during training?

[1] Holl, Philipp, Vladlen Koltun, and Nils Thuerey. "Physical Gradients for Deep Learning." arXiv preprint arXiv:2109.15048 (2021).

**Summary Of The Paper:**

This manuscript introduces a new optimization fashion bridging the Gauss-Newton method and the vanilla gradient descent method in the middle for physical optimization with neural networks. Traditionally, when a neural network is used for a physical problem, it will inevitably be affected by the unbalanced magnitudes dramatically. This manuscript proposes HIG, which is a middle point of two popular and individually advantageous optimization methods. It also provides a nice guideline on how to choose the hyperparameters and examine the efficacy of HIG on a concept-illustrating synthetic problem and three realistic physical problems.

**Summary Of The Review:**

This manuscript fills a blank for optimization in physical deep learning with a nice and clean method. The presentation is relatively clear to me. I would recommend an accept.

---

> ### Author Response · Authors · 2021-11-17
> **Response to Reviewer Khvk**
>
> Dear reviewer,
>
> Thank you for your review. We appreciate your interest in this research direction and are happy to answer your questions.
>
> 1) **Physical gradients**
> We think that our work and the physical gradients paper are complementary, and despite aiming for the same goal arrive at very different algorithms.
> While the physical gradients inherently require an inverse solver for the physics part, our work considers network and physics as an entity and half-inverts this as a whole.
> Our intuition here is that if an inversion is already necessary due to the involved physical processes, why not also include the network in the inversion to further improve results.
> It is worth mentioning that HIGs work on the level of linear approximations while physical gradients can use higher-order information of the physics part. As we already explained in the manuscript regarding the processing of derivatives of the single data points in a mini batch, HIGs perform a collective inversion of all data points. In contrast, gradient-based network optimizers compute an average of gradients over data points. Physical gradients process feedback from different data points by averaging as well. For future research, we are very interested to work on a more detailed analysis and comparison of the two approaches.
> 2) **Loss function of the toy example**
> The loss function in our submission has the scaling factor directly after the second output component of the neural network. This leads to a small entry in the neural network Jacobian simulating the effect of an ill-conditioned physics solver. Your suggested loss formulation, which contains the scaling factor after the difference of the second output component of the neural network and second component of the reference data, has this effect as well. However, it additionally rescales the reference data. As we intended to illustrate the different scales in the gradient flow for the same learning problem with the same reference data, we decided to use the first version in our paper.
> In order to describe motivation and our method in the second section in a consistent way, we needed a notation for different data points (subscript), different components (superscript), and the reference data (hat). This caused the definition of the loss of the toy example to be slightly more complicated than necessary. We clarified this in the text.
> 3) **Computational cost**
> It is correct that the computational cost is higher than for a "simple" gradient update. To provide specific numbers for the memory footprint for the non-linear oscillator example: For batch sizes of 32, 64, and 128, respectively, the corresponding memory requirements are 169 MB, 676 MB, and 2640 MB.
> Tensorflow allows for a partial evaluation of the Jacobians, saving memory at the expense of additional sequential calls.
> This is an effective way to reduce the memory requirements of our method in cases where the length of a HIG step is dominated by the half-inversion.
> Concerning potential benefits from an iterative method, this is a very good idea and we are currently actively exploring this direction.
> Since iterative solvers for both nonlinear least squares fitting and square root operations on matrices exist, a combination of both could potentially perform the half-inversion iteratively.
> Besides Krylov subspace methods, multigrid methods are also worth exploring in this context as the involved coarse-graining process is well-suited for the physical states in our experiments. The resulting advantages could be a computational speed-up as well as memory savings, as depending on the method, the computation of Jacobian-vector products or of the Jacobian on a given subspace could be sufficient.
> 4) **Usage of learning rate schedules**
> A learning rate schedule was not used for any of the experiments currently shown in the paper. We thought it was best to present the most elementary procedures for a clear comparison between the different approaches. Nevertheless, we conducted experiments with learning rate decay for the nonlinear oscillator example, which we will include as an additional baseline in the appendix. They show no significant change in the final performance of the trained models.
> 5) **Normalization layers**
> Normalization layers are not required for HIGs. As they only affect the neural network, it should also be mentioned that these do not affect or normalize the gradient flow through the physics solver. We did not use any normalization in the experiments presented in the paper but applying HIGs to normalization layers is straightforward: The additional trainable parameters appear like the other network weights as columns in the stacked Jacobian (eq. 6) and then the half-inversion is performed in the usual way.
> We conducted a first test with batch normalization layers that ran without problems. Training with or without normalization layers was equally stable, and both runs resulted in very similar final accuracies.

---

> > ### Comment · Reviewer_Khvk · 2021-11-29
> > **Thank you for the responses**
> >
> > Dear authors,
> >
> > Thank you for your responses. I think the authors have partially addressed my concerns. Vectorized and serialized back-propagation seem to have individual downsides in linear memory and time consumption, respectively. However, I guess it will not be a big problem here considering the performance improvement, and it sheds light on the future direction for some optimization. I would like to defend my score.

---

### Official Review · Reviewer_6S7S · 2021-11-02

**Correctness:** 3
**Technical Novelty And Significance:** 3
**Empirical Novelty And Significance:** 3
**Recommendation:** 6
**Confidence:** 3

**Main Review:**

The paper starts out with a very strong overview of past work, and cites Holl et al. (2021) for the motivation that existing optimizers do not perform well on joint optimization of NNs and physics. It would be very helpful if the paper was a little more self contained in the motivation. I see several references to "physics solvers" and "ill-conditioning" but no real motivation as to why these are related. The authors give an example of an ill-conditioned quadratic, which helps motivate why incorporating neural networks with ill-conditioned problems could lead to difficulty, but it's not clear why using "physics solvers" causes this.

Furthermore, I would like to see "physics solvers" to be defined a little bit more explicitly in the beginning of the paper. I was not able to quite understand until reading the appendix and all the case studies. I now assume that you mean a time-stepping scheme that you can differentiate through end-to-end (or using adjoint methods).

Section 2:
- It would be very helpful if you made vectors more explicit in the notation.
- e.g. In the "Physics optimization" subsection, you say "the sum reduces to a single data point". This confused me for a bit, until I realized $x$ and $y$ are vectors (but even then MSE would reduce to a sum of squared terms over the dimensions of the states).
- Equation (3) would be helpful to emphasize that you're using a pseudoinverse, not a standard inverse.
- It took me some time to understand that batch dimension is the number of points in the target state.
- Did you experiment with different configurations of $\kappa, \beta, \tau$? It would be interesting as ablation experiments to disentangle the benefits of these parameters. In particular, does most of the benefit come from $\beta$? Or is there also a benefit in setting $\kappa$ to an intermediate value?

Nonlinear oscillator:
- It might be useful for people who are less familiar to write out the actual time-stepping you get with Hamilton's equations. Maybe this can go in the appendix.
- How many Adam steps are you able to perform in one HIG step? It would also be useful to see number of steps required vs batch size.

Poisson problem:
- Correct me if I'm wrong, but it seems like this setup is different from the other ones in the sense that it's a surrogate training problem, rather than optimizing through a physics solver. Is that correct? I think it would be helpful to make this clear.
- The pretraining is interesting. Would GN perform well when pretrained with Adam?
- re: "the Poisson problem is relatively simple, requiring only a single matrix inversion": Arbitrarily ill-conditioned quadratics can also be solved by a single matrix inversion. I'm not convinced that is the reason for the observations.

**Summary Of The Paper:**

The paper proposes a way to interpolate between gradient descent and Gauss-Newton's method for solving nonlinear least squares problems arising from physics informed training. The authors cite past work that physics informed training is often ill conditioned, so gradient descent often performs poorly. They give a classic example of ill-conditioning, and show that gradient descent converges slowly, while Gauss-Newton quickly saturates neuron activations.

This motivates them to introduce Half-Inverse Gradients (HIG), which interpolates between GD and GN using the SVD. The authors then try the method on several test problems in scientific computing: Control of nonlinear oscillators, Poisson, and control of a Quantum dipole. They compare Adam, HIG, and GN.

**Summary Of The Review:**

The paper proposes a potentially interesting method to interpolate between GD and GN, and is certainly moving in the right direction of a very important problem in scientific machine learning, but the paper as it is right now is a little confusing to read. It would also benefit a lot from some ablation experiments showing the benefit of each individual part of their methods. I vote for marginally below threshold, but am willing to change my score if my concerns are addressed.

Let me know if you have questions about my review.

---

> ### Author Response · Authors · 2021-11-17
> **Response to Reviewer 6S7S - Part 2**
>
> * **Remarks on the Poisson problem**
>   * This problem indeed presents a surrogate training for a reconstruction problem. However, the way the neural network is trained is still by incorporating the physics solver between the network and the final L2 loss. So our setup does indeed include gradient flow through the Poisson solver.
>   * We conducted the experiment to use GN for the continued training run instead of HIGs.
> Like for HIGs, we observed a jump of the loss curve right after the start, but in contrast to HIGs, GN increases the loss values.
> Investigating why HIGs improve the loss so rapidly within a few mini batch updates, and also why, as shown in the appendix, a subsequent Adam training destroys this improvement again, will be a very interesting avenue of future research.
>   * Regarding the "simplicity of the single matrix inversion" mentioned in section 3.2 of our paper, we wanted to highlight that this combination of network-physics and optimization task is non-convex, but with the non-convexity arising only from the neural network. The physics solver is only a linear map, which, of course, is still ill-conditioned. For the non-convexity of neural networks, Adam is a well-tuned and established optimizer.
> Our experiments indicate that this at least partially accounts for the reduced improvements from training with HIGs compared ot the other cases.
> We are aware that this is not a strictly formal argument, but it nonetheless provides some intuition for the observed behavior.
>
> * **Clarifications for the motivation section of our paper**
> The terms "physics solver" and "ill-conditioned" are related to how a given physical system with its high-frequency modes is translated into the language of numerics. In the process, the physical system is mathematically modeled by a partial differential equation and afterward, differential operations are discretized to receive a numerical scheme that can be used on a computer. This scheme is what we call a physics solver and can correspond to a single forward solve (Poisson example) or to many time integration steps (oscillator and quantum dipole examples). Differentiability of such schemes is mathematically always possible, but not always given for existing implementations.
> Nonetheless, it is a requirement for the purposes of our paper. Sensitive behavior of the physical system connected to its high-frequency modes is incorporated in the physical solver and leads to small singular values in its linear approximation, which yields an ill-conditioned Jacobian matrix.
> The ill-conditioned nature can be illustrated with a well-known example: Consider the spectrum of the discretized Laplace operator in 1D with N discretization points. With larger N, this matrix is increasingly ill-conditioned. This stems from small eigenvalues occurring due to the physical states on finer spatial scales being modeled with an increased number of discretization points N. From a physical viewpoint of Poisson's equation in electrostatics, this is the mathematical equivalent of our observations that the potential field of two electric point charges changes abruptly when bringing them close together.
> We will include this explanation in the paper in a condensed form to help readers better understand our motivation.

---

> ### Author Response · Authors · 2021-11-17
> **Response to Reviewer 6S7S - Part 1**
>
> Dear reviewer,
>
> Thank you for your review. We appreciate your detailed suggestions where our manuscript needs further clarification and are happy to answer your questions.
>
> * **Remarks on section 2**
>   * We will follow your suggestions to use vector notation in our expressions, clarify in the text that the sum over vector components is part of  equation 2, and point out that we use the pseudo-inverse in equation 3.
> The batch dimension refers to the number of data points in a mini batch used for one update step.
> We denote the number of points in the target state as the dimension of the physical state, which corresponds to the number of discretization points. Batch size multiplied by the dimension of the physical state gives the number of rows of the stacked Jacobian, which is required for a HIG update.
>   * Concerning your suggestion of an ablation study, we conducted several experiments with different settings on the mentioned parameters for the nonlinear oscillator system.
> We will include the corresponding plots and discussion in the appendix of our paper, and summarize them below.
> __$\beta$ (exponent of scalar prefactor):__
> We started simulations for different values $\beta$ between $0.0$ and $-0.5$. The loss curves are stable and very similar. Final loss values are between $2.8\cdot 10^{-6}$ and $4.9\cdot 10^{-6}$. Compared to the other parameters, $\beta$ has only little influence on the result.
> __$\tau$ (truncation parameter for singular value decomposition):__
> Truncation can be used to stabilize the inversion. We chose it according to our experiences from how large $\tau$ needs to be in order to solve the pure physics optimization task without neural network. For the nonlinear oscillator system,
> we conducted further experiments with different $\tau$ from $10^{-6}$ to $10^{-10}$. The HIG curves start to converge when lowering the truncation, with final loss values as follows:
> | Method        | Truncation parameter $\tau$          |   Final loss value  |
> | ------------- |:-------------:|:-----:|
> | HIG     | $10^{-6}$ | $3.3\cdot 10^{-6}$ |
> | HIG      | $10^{-8}$     |   $2.5\cdot 10^{-6}$ |
> | HIG | $10^{-10}$      |     $2.2\cdot 10^{-6}$|
>
>     Thus, across a wide range of choices for $\tau$, our method converges to a low final loss value. Please also see the response to reviewer itKt for more aspects regarding the $\tau$-parameter.
> __$\kappa$ (exponent of diagonal singular value matrix):__
> We tested $\kappa$ for various values between $1$ (SGD) and $-1$ (GN). For positive values, curves stagnate early with final loss values around $10^{-3}$. Negative values for $\beta$ produce results several orders of magnitude better around $10^{-6}$. The HIG curve achieves the best result and stagnates below $10^{-7}$. This supports our argumentation that a strong dependence on this parameter exists, and that a choice of $\kappa=-0.5$ is indeed a good compromise for scale-correcting updates of reasonable size.
> The strong improvement as soon as $\kappa$ becomes negative indicates that the collective inversion of the feedback of different data points in the mini batch is an important ingredient in our method.
> | Method        | Exponent $\kappa$          |   Final loss value  |
> | ------------- |:-------------:|:-----:|
> | SGD (no Jacobian required)     | 1 | $6.7*10^{-4}$ |
> |       | 0.5     |   $1.1*10^{-3}$ |
> |  | 0     |     $2.5*10^{-3}$|
> |      | -0.25 | $4.6*10^{-7}$ |
> | HIG      | -0.5     |   $6.3*10^{-8}$ |
> |  | -0.75      |     $4.6*10^{-7}$|
> | GN | -1      |     $1.0*10^{-6}$|
>
>
> * **Remarks on the nonlinear oscillator experiments**
>   * We agree that it is a good idea to give the explicit form of the differential equations and will do so in the appendix. The reason for the Hamiltonian formulation was that the notation is more compact.
>   * The number of Adam steps performed in one HIG step depends on the batch size. Below, we list the average duration per mini batch step in seconds:
> | Batch size        | Adam step duration         |   HIG step duration |
> | ------------- |:-------------:|:-----:|
> | 32     | 0.081 | 0.087 |
> | 64      | 0.081     |   0.097 |
> | 128 | 0.081      |     0.146 |

---

### Author Response · Authors · 2021-11-17
**General Response to All Reviewers**

Dear reviewers,

Thank you for your feedback. We are very happy to see that all of you could follow our ideas and argumentation. We will update our manuscript with respect to all of your suggestions where formulations, notation, or the argumentation was found to be confusing.

Below, we address your questions in the individual responses. We want to express our gratitude for your very helpful feedback, and hope that you enjoyed reading our work.

---

### Author Response · Authors · 2021-11-22
**Updated draft**

Dear reviewers,

We have just posted the updated draft of our work with the changes that we announced in the responses to your feedback. These updates are marked with blue color in the paper. Besides some minor changes in formulation and layout, our new draft includes:

1) An ablation study for the hyperparameters of our method. This ablation study is attached as the last section of the appendix and referenced in the main text.

2) In section 2 (Physics Optimization), we explain how the terms ''physics solver'' and ''ill-conditioned'' optimization are related.

3) Regarding the computational cost, we give further details on the memory footprint. In the appendix B (Experimental Details), we list concrete numbers for memory requirements and the duration of the update step  of our method in comparison with Adam.

---

### Decision · Program_Chairs · 2022-01-20

**Decision:**

Accept (Spotlight)

**Comment:**

Thank you for your submission to ICLR.  The reviewers and I are in agreement that the paper presents a substantial contribution to the field at the intersection of differentiable simulation and ML methods.  In particular, the half-inverse method is compelling, non-obvious, and hints of a nice path forward towards the goal of practical differentiable simulations within models.  Overall I'm happy to recommend the paper be accepted.